| Open Peer Review | Computational Biology | Methods and Protocols

# GEMsembler: consensus model assembly and structural comparison of genome-scale metabolic models across tools improve functional performance

Elena K. Matveishina,[1,2] Bartosz J. Bartmanski,[1] Sara Benito-Vaquerizo,[1] Maria Zimmermann-Kogadeeva[1]

**ABSTRACT** Genome-scale metabolic models (GEMs) are widely used in systems biology to investigate metabolism and predict perturbation responses. Automatic GEM reconstruction tools generate GEMs with different properties and predictive capacities for the same organism. Since different models can excel at different tasks, combining them can increase metabolic network certainty and enhance model performance. Here, we introduce GEMsembler, a Python package designed to compare cross-tool GEMs, track the origin of model features, and build consensus models containing any subset of the input models. GEMsembler provides comprehensive analysis functionality, including identification and visualization of biosynthesis pathways, growth assessment, and an agreement-based curation workflow. GEMsembler-curated consensus models built from four *Lactiplantibacillus plantarum* and *Escherichia coli* automatically reconstructed models outperform the gold-standard models in auxotrophy and gene essentiality predictions. Optimizing gene-protein-reaction (GPR) combinations from consensus models improves gene essentiality predictions, even in the manually curated gold-standard models. GEMsembler explains model performance by highlighting relevant metabolic pathways and GPR alternatives, informing experiments to resolve model uncertainty. Thus, GEMsembler facilitates building more accurate and biologically informed metabolic models for systems biology applications.

**IMPORTANCE** Genome-scale metabolic models (GEMs) capture our knowledge of cellular metabolism as encoded in the genome, enabling us to describe and predict how cells function under different conditions. While several automated tools can generate these models directly from genome data, the resulting models often contain gaps and uncertainties, highlighting areas where our metabolic knowledge is incomplete. Here, we introduce a new tool called GEMsembler, which integrates GEMs constructed by different methods, evaluate model uncertainty, and build consensus models, harnessing the unique features of each approach. These consensus models more accurately reflect experimentally observed metabolic traits, such as nutrient requirements and condition-specific gene essentiality. GEMsembler facilitates comprehensive analysis of model structure and function, helping to pinpoint knowledge gaps and prioritize experiments to address them. By synthesizing information from diverse sources, GEMsembler accelerates the development of more reliable and biologically meaningful models, advancing research in metabolic engineering, pathogen biology, and microbial community studies.

**KEYWORDS** metabolic modeling, genetic algorithm, *Escherichia coli*, *Lactobacillus*

G enome-scale metabolic models (GEMs) are among the fundamental tools in systems biology used to describe cellular metabolism and predict perturbation responses

**Peer Reviewer** Daniel Machado, Norges Teknisk-Naturvitenskapelige Universitet, Trondheim, Norway

Address correspondence to Maria Zimmermann-Kogadeeva, maria.zimmermann@embl.de.

The authors declare no conflict of interest.

See the funding table on p. 24.

(1, 2). GEMs are reconstructed based on genome annotations and represent a metabolic network of reactions and metabolites associated with enzymes via gene-protein-reaction (GPR) rules. Flux balance analysis (FBA) and its variations are used to estimate metabolic fluxes in the network under given conditions, thus predicting growth and consumption and production of metabolites, while allowing integration of different types of experimental data to constrain the model (3, 4). The model quality is therefore crucial for accurate model predictions (5).

Although manual curation remains the gold standard for production of high-quality models (5), there are various tools for the automatic reconstruction of draft GEMs that utilize different approaches and can be used as a starting point (6). Some tools, like gapseq (7) or modelSEED (8, 9), follow a bottom-up approach by mapping enzyme genes found in the genome to the known reactions from the biochemical databases and subsequently filling the gaps to form a complete network. An alternative top-down approach is proposed in the CarveMe tool (10), which starts with a universal model from the BiGG (11) database and carves out unnecessary reactions based on the enzyme presence. For human gut bacteria, there is a widely used AGORA collection of semi-automatically built models (12, 13), which can be downloaded from the Virtual Metabolic Human database (14). Each GEM reconstruction tool has its advantages, and none of the tools consistently outperforms the others (6).

GEMs built by different tools for the same organism often use different database nomenclatures, have different structures and functional performance, making a direct comparison challenging (3). For example, modelSEED models are built using the modelSEED database (8), gapseq relies on several integrated databases, including ModelSEED (8) and MetaCyc (15), while CarveMe uses the BiGG database (11). One approach for unifying nomenclature is offered by MetaNetX (16, 17), an online platform that connects metabolites and reactions namespaces from different databases. However, while comparing lists of metabolites and reactions provides an overview of models' similarities, the structural and functional differences between GEMs are not revealed. Alternatively, one can compare models based on their functional performance, such as prediction of growth, auxotrophy, or gene essentiality, compared to the experimental data (18–22). However, this approach does not reveal the differences in the network structures of models constructed with different tools.

Emerging cross-tool studies (23–25) show that models built with different tools can capture various aspects of metabolic behavior, and therefore combining them in one model may improve the model performance. Several frameworks were proposed to merge GEMs built with different tools. For example, modelBorgifier (26) allows merging of two models in a semi-automated manner, while mergem (23) automatically generates a union model from several input GEMs containing metabolites and reactions from the original models. However, to date, no framework can merge all model features, including genes and GPRs, track the origin of each feature in the output model, and generate fine-tuned, flexible combinations of GEMs.

To address this need, we developed GEMsembler, a Python package for comparing, combining, and analyzing GEMs built with different tools. GEMsembler has the following unique features: (i) it enables structural comparison of GEMs built with different tools; (ii) it systematically assesses confidence of the metabolic network at the level of metabolites, reactions, and genes; and (iii) it provides a comprehensive framework for assembling different combinations of the input models and assessing their predictive capacity in terms of growth, auxotrophy, and gene essentiality. Furthermore, consensus models generated from the input GEMs can be curated using the GEMsembler functionality in a semi-automated manner. Using the two model organisms *Escherichia coli* and *Lactiplantibacillus plantarum* (formerly *Lactobacillus plantarum*) as examples, we demonstrate that GEMsembler-curated consensus models can outperform the current gold-standard manually curated models in auxotrophy and gene essentiality predictions. Furthermore, the GEMsembler framework highlights the features that explain the improved model

performance, thus providing valuable information for targeted experimental validation to elucidate the knowledge gaps and uncertainties in metabolic networks.

## RESULTS

### Generating cross-tool consensus models with GEMsembler

GEMsembler assembles GEMs following a workflow consisting of four major steps: (i) conversion of features of the input models (metabolites, reactions, and genes) to the same nomenclature, (ii) combination of the converted input models into one object, which we refer to as supermodel, (iii) generation of consensus models containing different combinations of the input models' features, and (iv) comparison and analysis of the consensus models (Fig. 1A; Fig. S1).

First, GEMsembler converts metabolite IDs of the input models to BiGG IDs (11) using different sources of information linking IDs from various databases and BiGG. Next, converted metabolites are used to convert reactions to BiGG nomenclature via reaction equations (Fig. S1) to ensure that the converted model maintains the same topology as in the original models. Finally, if genome sequences are provided along with the input models, GEMsembler converts genes from the input models to the locus tags of a genome selected as the output using BLAST (27) (Fig. S1). GEMsembler keeps track of all the intermediate conversion steps, providing the possibility to troubleshoot in case of conversion issues.

After performing the conversion, GEMsembler assembles all converted models into one supermodel (Fig. 1A; Fig. S1). The supermodel follows the structure of the COBRApy Python class (28), while having additional fields to store the information about the converted features (metabolites, reactions, or genes) and their source of origin (Fig. 1B; Fig. S1). Features that could not be converted are stored in a separate field (termed "not_converted") in the supermodel. During the creation of the supermodel, only the union of the input models (termed "assembly") is generated, which includes all features present in at least one model. All the other possible combinations of the input models, termed "consensus" models, are generated in the next workflow step. For example, we can generate "coreX" consensus models with features that are included in at least X of the input models (assembly is therefore identical to the core1 model). We define the feature confidence level as the number of input models that include this feature. The feature attributes in the consensus models are assigned according to the same agreement principle. For example, if a reaction is unidirectional in three out of four input models and bidirectional in one model, it will be unidirectional in core4, core3, and core2 consensus models and bidirectional in the assembly model. GPR rule attributes of reactions are compared based on the logical expressions involving genes from the original models to create new GPRs for the output consensus models. The consensus models are stored as a field in the supermodel object and can be extracted as separate models in the standard SBML format for downstream analysis with COBRA tools, such as flux balance analysis, gene essentiality prediction, and other functions.

### Investigating the structure and functions of the metabolic network with consensus models

The GEMsembler supermodel consists of metabolites, reactions, and gene objects, which can be examined interactively, and provides information about the original models' agreement on the network features, which can be used to assess their confidence and identify gaps that could be experimentally validated. Due to the large number of reactions, metabolites, and genes present in GEMs, it is challenging to identify subnetworks or features of particular interest for further exploration. To facilitate this analysis, we integrated additional functionality in GEMsembler (Fig. 1C) to systematically investigate the structure and functions of the metabolic network and identify subnetworks with various levels of confidence using consensus models.

To explore the network structure, GEMsembler implements neighborhood search to identify all reactions at a given distance from a metabolite of interest, as well as three

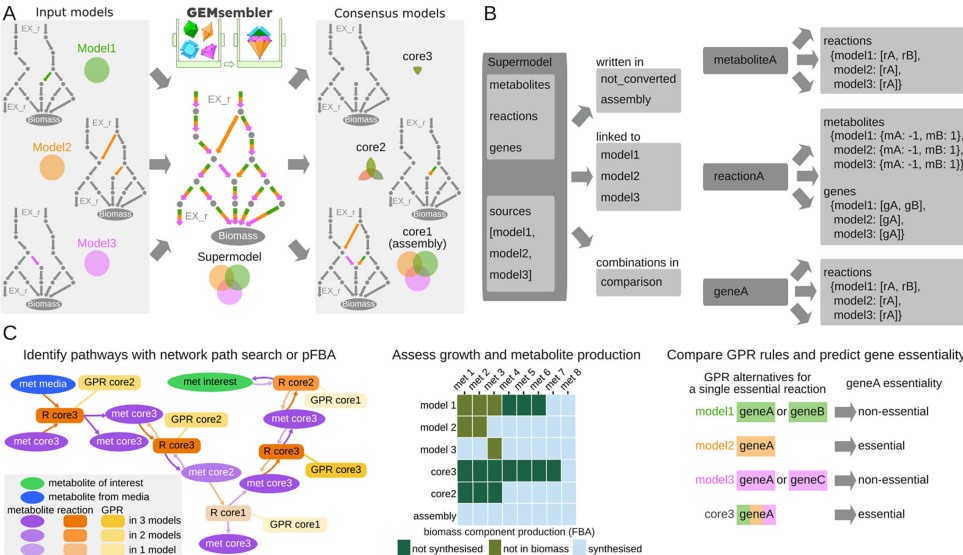

**FIG 1** GEMsembler converts metabolic networks built with different tools to one nomenclature, combines them into a supermodel, and offers diverse comparison functionality. (A) Schematic representation of GEMsembler workflow, including construction of supermodel and generation of consensus models with different confidence levels. (B) Supermodel structure resembles the structure of a COBRApy model Python class with additional information on feature conversion and their original sources. (C) Examples of the downstream analysis functionality, including pathway visualization (left), growth analysis by assessment of metabolite production (center), and testing gene essentiality via GPR combination (right).

ways of defining a pathway, for which the confidence should be assessed. First, the pathway can be predefined by the user (e.g., glycolysis, tricarboxylic acid [TCA] cycle). Second, the pathway can be calculated from the network by identifying all possible ways of synthesizing metabolites of interest in a given culture medium using a topology approach. In this approach, all possible paths (sequences of reactions) between the input medium components and the target metabolites are calculated by the MetQuest package (29) integrated into GEMsembler. This topology approach generates pathways that resemble classically defined biochemistry pathways, for example, in the KEGG (30) or MetaCyc (15) databases. While these pathways exist in the network, it is not guaranteed that they can carry flux, as this approach does not perform flux analysis at the network scale. Therefore, the third approach to determine the biosynthesis pathways is to perform parsimonious flux balance analysis (pFBA) (31), optimizing the production of each metabolite of interest while identifying the solution with the minimal flux. Once the pathway is defined, its confidence can be assessed by comparing the agreement scores for its metabolites, reactions, and the corresponding GPR rules between the input models. This information is provided as an output table and as an interactive pathway map that can be explored visually (Fig. 1C).

To investigate the model functions, GEMsembler assesses the models' ability to grow in a given medium by performing FBA using biomass production as the objective function. Next, GEMsembler performs FBA using the production of each component of the biomass reaction as the objective function to identify which metabolites cannot be produced in case the models are not able to grow (Fig. 1C). Finally, GEMsembler incorporates GPR rules predicted by the different models and assembles their combinations. Having several GPR options per reaction can be used to guide model curation with respect to the gene essentiality prediction (Fig. 1C).

## GEMsembler enables systematic characterization of uncertainties in GEMs

To demonstrate GEMsembler functionality, we investigated GEMs of two well-studied bacteria: *Lactiplantibacillus plantarum* (formerly *Lactobacillus plantarum*) WCFS1 (LP),

a gram-positive bacterium living in fermented foods and the gastrointestinal tract, auxotrophic to multiple nutrients, and *Escherichia coli* BW25113 (EC), the best studied gram-negative bacterium with a large collection of gene essentiality data. Since both of these organisms have manually curated GEMs (32, 33), we can use them as gold standards for comparison with the outcomes of the GEMsembler workflow. We set out to (i) automatically generate GEMs for the two organisms with three different tools: CarveMe, gapseq, modelSEED, and download the corresponding models from the AGORA collection; (ii) assemble supermodels with GEMsembler; (iii) assess the agreement of metabolic reactions and GPRs by investigating different consensus models: from the union of all models (assembly) to the intersection of all models (core4—since we are investigating four input models per bacteria); and (iv) compare the consensus models to the gold-standard ones.

The majority of metabolites, reactions, and genes from the four original models were successfully converted and included in the supermodels (Fig. S2; Table S1). Approximately half of the metabolites and reactions were identified only by one model, while between a quarter and a third of the genes were included in the GPR rule by only one model (LP: 339 out of 1186 genes; EC: 663 out of 1952 genes). Complete agreement between all four models was observed for no more than a quarter of metabolites, reactions, and genes. For each reaction with a GPR rule, we also calculated GPR agreement, which often does not correspond to the agreement score of the reaction itself. In general, the agreement between *E. coli* models is higher than between *L. plantarum* models (Fig. S2; Table S1).

To investigate agreement between different pathways, we focused on the production of central carbon metabolites (Fig. 2A and B, Table S2), and biomass components (metabolites included in the biomass reaction) (Fig. S3 and 4; Table S3). We first identified topologically possible biosynthesis pathways in PMM5 minimal medium reported for *L. plantarum* (34) and in M9 minimal medium for *E. coli*. Next, we assessed the confidence of each identified pathway by checking the confidence of reactions and GPRs in each path. We found that for most of the central carbon metabolites, the models unanimously agree on whether they are produced or not. All four *L. plantarum* models also agree on the way most of the central carbon metabolites are produced, although the GPR agreement is lower (Fig. 2A). For *E. coli,* there is absolute agreement between models for all cases except one, beta-D-glucose 6-phosphate, which includes several reactions present in three models out of four (Fig. 2B). While for all other metabolites, there is core4 agreement for GPRs for almost all reactions, four reactions have GPR with core3 agreement: HEX1, EDA, and ALKP are involved in the synthesis paths of most of the tested metabolites, and FBA3 is involved in the synthesis of D-Fructose 1,6-bisphosphate (Fig. 2B; Table S2). We also assessed the confidence of canonically defined glycolysis, the pentose phosphate pathway, and the TCA cycle. As expected, there is almost complete agreement in the glycolysis pathway with only some discrepancies on the GPR level (Table S4; Files S1 and S2). The pentose phosphate pathway is also consistent with disagreement in one *L. plantarum* reaction and three *E. coli* reactions out of eleven (Table S4; Files S3 and S4). The TCA cycle showed the most uncertainty: for *E. coli,* mostly on the GPR level, but for *L. plantarum,* most of the TCA cycle is either absent or identified with large discrepancies (Table S4; Files S5 and S6).

We next performed the same topology-based analysis for the production of biomass components. Different models include different metabolites in the biomass reaction; here, we decided to use their union for the complete overview (76 reactants in the union of both bacteria) (Table S5). From this analysis, we excluded metabolites that are either media components or cofactors. Production of biomass components is much less confident than that of central carbon metabolites for both organisms, but the trend that *E. coli* models have more agreement stands. For example, only 6 biomass components in the *L. plantarum* model were produced with complete agreement, while 11 biomass components had core4 agreement in *E. coli* (Fig. S3 and 4; Table S3).

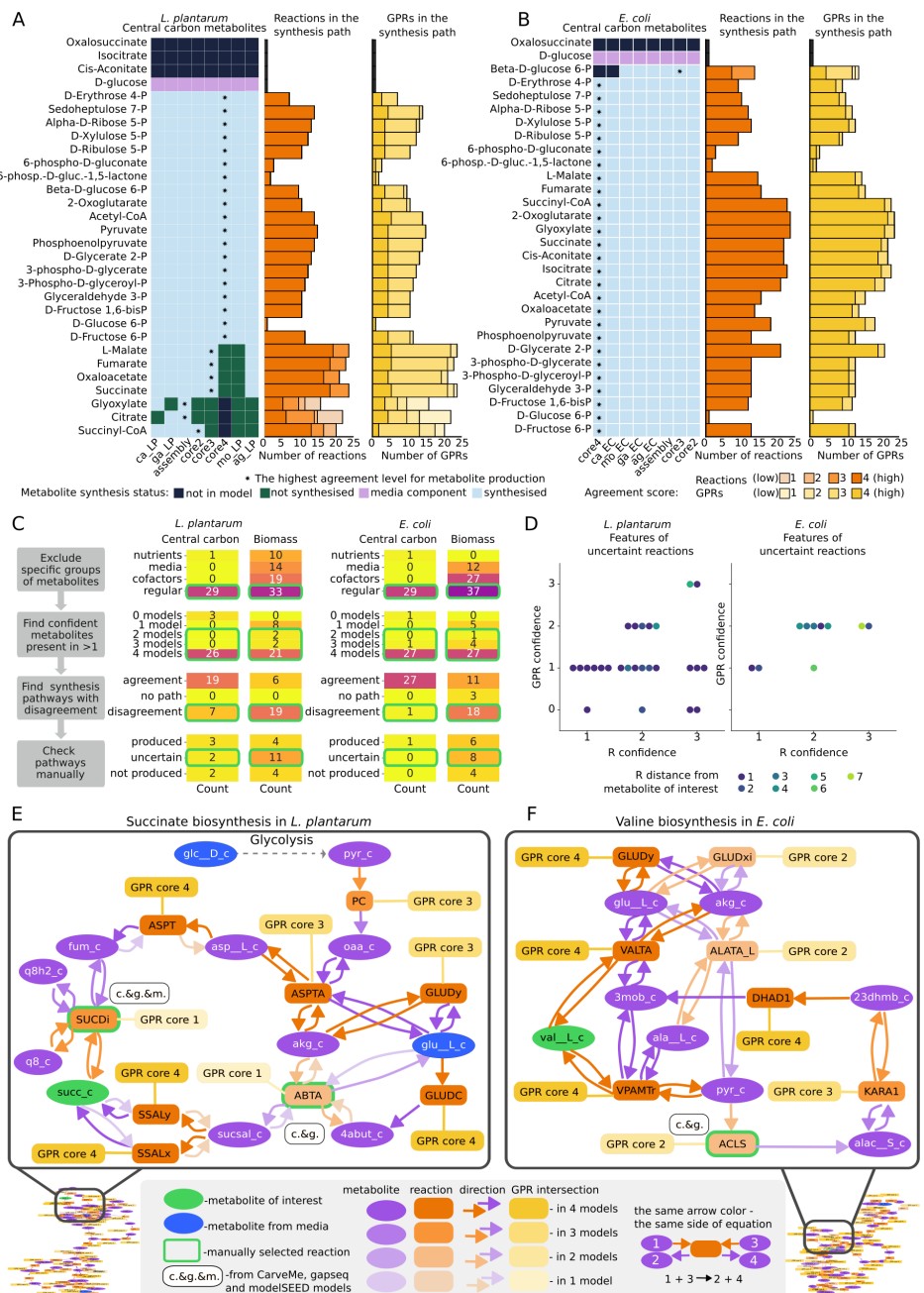

FIG 2 Analysis of metabolic network confidence for *L. plantarum* and *E. coli* models. (A) Central carbon metabolites production in the converted original models and consensus models on the left, and agreement scores for reactions and GPRs in the corresponding most confident pathways (highlighted with *) for *L. plantarum*. (B) The same plots for *E. coli*. (C) Step-wise procedure of identification of the most uncertain pathways and reactions for further investigation (left) and the results of central carbon metabolites and biomass components examination in *L. plantarum* and *E. coli* (right). (D) Characteristics of the most unconfident reactions selected at the last step in C for *L. plantarum* and *E. coli* with respect to their confidence, confidence of the corresponding GPR rules, and the distance from the selected metabolite of interest (which requires these reactions to be synthesized). (E) Example of an uncertain pathway in *L. plantarum* (succinate biosynthesis, caused by the SUCDi and ABTA reactions). (F) Example of an uncertain pathway in *E. coli* (valine biosynthesis, caused by the ACLS reaction).

Summarizing the results of the topology analysis allows us to identify areas of uncertainty in the large metabolic network, which should be further investigated in a step-wise approach (Fig. 2C). First, for the metabolites of interest (30 central carbon and

76 biomass components), we separated the regular metabolites from the other types of metabolites, such as nutrients, media components (other than nutrients), and cofactors. Next, for regular metabolites, we assessed the model agreement score and filtered out metabolites below the agreement score core2. This left us with 51 metabolites for *L. plantarum* and 60 for the *E. coli* model, for which we then checked production pathways. Metabolites without complete model agreement on their biosynthesis paths were then subject to manual examination (Fig. 2C; Table S6).

In this final step, we examined the biosynthesis pathways of 7 central carbon metabolites and 19 biomass components of *L. plantarum*, as well as 1 central carbon metabolite and 18 biomass components of *E. coli* using the interactive maps generated by GEMsembler and manually assigned them into three categories representing our decision about their production: produced, not produced, and uncertain (Fig. 2C; Table S6). The pathways from the uncertain category contain reactions that are needed to produce metabolites of interest and that have discrepancies between two or more models on the gene level. Therefore, the disagreement between models does not make us more certain about whether the metabolites can be produced and the corresponding pathways should be prioritized for further investigation. For *L. plantarum*, these include pathways for 2 central carbon metabolites, succinate and succinyl-CoA, and 11 biomass components, while for *E. coli,* these include 8 biomass components (Fig. 2C; Table S6). Each of the uncertain biosynthesis pathways contains at least one reaction causing the uncertainty, with a total of 26 uncertain reactions for *L. plantarum* and 10 uncertain reactions for *E. coli* (Fig. 2D; Table S6). Most of these reactions in *L. plantarum* are confirmed by two models and have GPR included only in one model, while for *E. coli*, uncertain reactions have slightly better confidence with GPRs included in two models. Most of these uncertain reactions are in the immediate proximity to or one reaction away from the investigated metabolite, while some are up to seven reactions away in the biosynthesis pathway (Fig. 2D; Table S6).

GEMsembler provides the opportunity to explore these pathways and uncertainties visually with interactive maps and select their key elements, as we show for succinate biosynthesis in *L. plantarum* (Fig. 2E; Table S6; File S7). In this case, uncertainty is caused by two alternative paths: one through the SUCDi reaction and another through the ABTA reaction with GPRs provided only by one model. Another example is valine biosynthesis in *E. coli* with three reactions found only by two models: ACLS, GLUDxi, and ALATA_L (Fig. 2F; Table S6; File S8). GLUDxi duplicates the function of a more confident GLUDy, with the only difference being the use of NAD instead of NADP and, therefore, does not influence valine production. Since glutamate dehydrogenase in *E. coli* is NADP-specific (35, 36), this reaction is likely erroneously added by the automatic reconstruction tools. ALATA_L is required for alanine biosynthesis, and therefore was considered an uncertainty in the alanine pathway. It has been reported that *E. coli* encodes at least two alanine-synthesizing glutamic-pyruvic transaminases (37); therefore, this reaction is likely correctly added to the model. The ACLS reaction corresponding to acetolactate synthase is directly required to produce valine and has been identified in *E. coli* (38, 39); therefore, it should also be retained in the model. In this way, identified uncertain reactions can either be validated by existing knowledge or serve as candidates for further experimental verification.

In this part, we demonstrated how GEMsembler helps to systematically characterize the confidence in the metabolic network and prioritize uncertain pathways and reactions for further investigation based on their confidence levels.

## Curation of GEMs with GEMsembler to reproduce growth phenotypes

The consensus models constructed with GEMsembler can facilitate model curation to reproduce growth with the classical FBA algorithm. Since both *L. plantarum* and *E. coli* can grow in defined minimal media (Table S7), we can use this information to curate their models with GEMsembler following the agreement principle. GEMsembler growth analysis functionality uses FBA to identify biomass components that cannot be

produced, therefore explaining the lack of growth under the given conditions. Preliminary growth simulations for mixed original and consensus models demonstrated that neither of the models could grow, because not all of the biomass components could be produced (Table S8).

To curate the models, we decided to first modify the biomass reaction and then make sure that all biomass components can be produced with FBA. We used the agreement score calculated by the GEMsembler biomass analysis function to keep biomass components included by three or more models (Table S5). Several metabolites included only by one or two models were also included if they could be synthesized by the corresponding models (Materials and Methods). In total, we included 58 and 61 components in the biomass reactions of *L. plantarum* and *E. coli*, respectively (Table S5).

After modifying the biomass reaction based on the agreement principle, we ran FBA for all models again to test whether they gained the ability to grow. Only the assembly model was able to grow, while each of the other models was not able to produce at least one biomass component (Fig. 3A and B; Table S8). Here, we used FBA instead of topology analysis to ensure that the metabolite production is relevant for the growth simulations and takes into account transport and cofactor utilization. The number of metabolites produced with complete agreement (core4) was low in both organisms (LP: 10/58; EC: 12/61), but *E. coli* can produce more biomass components with higher agreement than *L. plantarum*. For *E. coli*, the core2 consensus model can produce the majority of biomass precursors (Fig. 3A), while for *L. plantarum,* core2 and core3 consensus models were similar in their metabolite production capacity and could produce less than half of the target metabolites (Fig. 3B), underlining that model consistency is much lower for *L. plantarum* compared to *E. coli*. We note that there are metabolites (e.g., cysteine, glutathione, 2-demethylmenaquinone) that can be produced by all input models, but not by all consensus models (Fig. 3A), which indicates that the production pathways are different between the models and are therefore not included in the consensus. To balance the confidence, complexity, and function (being able to grow) of the curated model, we chose the core3 consensus model as the basis and added a subset of reactions from the other models to ensure that all biomass precursors can be produced.

To determine which reactions are necessary to add for curation, we examined the interactive maps depicting pFBA-determined biosynthesis pathways for biomass components that cannot be produced by the core3 model, but can be produced by some other models. Reactions with less than core3 confidence level in these pathways from the other models can restore the production of the target metabolite, even if they are not in its immediate proximity, but at the same time, not all of them are essential for production. For example, in the path for thiamine diphosphate biosynthesis in *L. plantarum*, there are two to three unconfirmed reactions in three parts of the pathway that would need to be added (Fig. 3C; Table S8; File S9). One of the most challenging parts of curation is considering the highly connected biomass precursors such as ATP or NAD. In this case, sometimes adding reactions will not be enough, and reaction properties need to be changed. For example, the CarveMe model was the only one able to produce ATP in *L. plantarum*, which was due to the bidirectionality of the phosphoribosylaminoimidazole carboxylase reaction (AIRCr); therefore, we implemented this bidirectionality in the curated core3 model as well.

Overall, we added 72 reactions to the core3 *L. plantarum* model, including 28 transport and exchange reactions, leading to the final core3 GEMsembler curated model with 639 metabolites, 729 reactions, and 420 genes. We curated the core3 *E. coli* model in the same way to ensure its growth on M9 minimal medium. We added 43 reactions, including 11 for transport and exchange, and the final core3 GEMsembler curated *E. coli* model consists of 943 metabolites, 1,217 reactions, and 644 genes. The technical quality reports generated by MEMOTE (40) for the final core3 models and the original models confirm that GEMsembler curation solves the issue of blocked biomass precursors that exists in all original models except CarveMe and leads to a more realistic growth for *L. plantarum*.

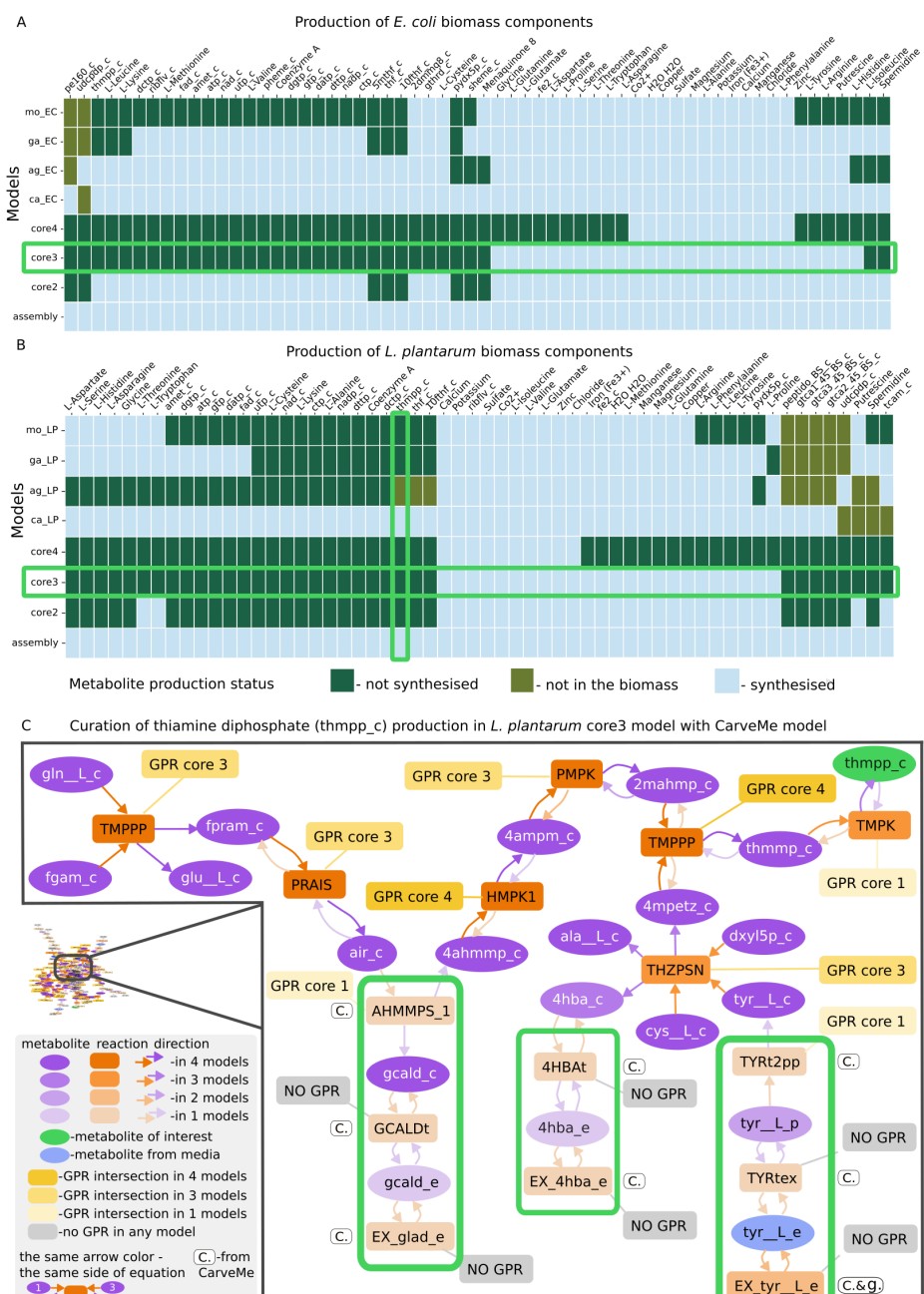

**FIG 3** Curating *L. plantarum* and *E. coli* models with GEMsembler. (A) Production of unified biomass precursors for *L. plantarum* estimated with pFBA. (B) Production of unified biomass precursors for *E. coli* estimated with pFBA. Horizontal green boxes highlight the models selected for curation; the vertical green box highlights the metabolite for which the curated pathway is depicted in panel C. (C) Example of curated reactions in thiamine diphosphate (tmpp_c) biosynthesis pathway of *L. plantarum*.

In this section, we demonstrated how GEMsembler can aid model curation in a semi-automated way to ensure that the model can reproduce a known growth phenotype.

# GEMsembler-curated model outperforms the gold-standard *L. plantarum* model in auxotrophy prediction

After curating the core3 model of *L. plantarum* for growth in PMM5 minimal medium, which contains several amino acids and vitamins, we wanted to assess the model quality by testing whether it can predict *L. plantarum* auxotrophy to different nutrients. We compared the core3 model's topology and performance to a previously reported curated model of *L. plantarum* iLP728 (6, 32), which we considered to be the gold standard, and to the four original AGORA, CarveMe, gapseq, and modelSEED models of *L. plantarum,* converted to BiGG nomenclature with GEMsembler and automatically gap-filled with the CarveMe tool on the minimal PMM5 medium, which we considered to be the baseline. We compared how many reactions and genes intersect between each of the models and the gold-standard iLP728 model and calculated the ratio of each intersection to each of the model's size (precision) or the iLP728 size (recall) (Fig. 4A; Table S9). We also calculated the F1 score (the harmonic mean of precision and recall) as a summary metric of the reaction and gene overlap. Compared to the original input models, the core3 model recalls a similar fraction of reactions and genes corresponding to these reactions in the iLP728, while including much fewer reactions and genes not present in the iLP728 model, leading to the best F1-score for reactions (0.58) and second best F1-score for genes (0.51) (Fig. 4A; Table S9). Thus, the GEMsembler curated core3 model is more similar to the gold-standard iLP728 model of *L. plantarum* than the four original models.

While higher model similarity to the gold-standard iLP728 does not necessarily mean higher model quality, as new gene functions and pathways could have been discovered since the original publication of iLP728, we next assessed the models' functional performance. We tested the models' ability to predict *L. plantarum* auxotrophy to different nutrients in a different medium, CDPM (Table S7), and compared the results to the reported experimental auxotrophy data classified into three categories: growth, no growth, and reduced growth (32, 34). We predicted growth for each model in each condition by running FBA in a modified medium where one of the tested nutrients was removed. We classified FBA prediction results into growth and no growth since there were no cases with reduced growth prediction. Our core3 model outperforms all the other models, including iLP728, in two to four growth conditions. There are no metabolites for which auxotrophy prediction is incorrect in the core3 model but correct in any other model (Fig. 4B; Table S9). Four out of five false predictions of the core3 model happen in the case of experimentally observed reduced growth, while the model predicts either growth or no growth. In the case of tryptophan auxotrophy, where the core3 and all other models predict growth, the false prediction can likely be explained by the non-metabolic inhibition by other aromatic amino acids in the medium (32). Compared to the gold-standard iLP728 model, core3 correctly predicts auxotrophy phenotype for four metabolites: growth without biotin and pyridoxamine, and absence of growth without glutamate and riboflavin (Fig. 4B; Table S9).

The GEMsembler functionality to analyze the network structure allows us to investigate the underlying reasons for improved auxotrophy predictions. In the case of biotin, the difference is that the iLP728 model considers it as a biomass component, while none of the original models, and therefore neither the core3 model includes it (Table S5). Although this modification of the biomass reaction leads to the correct prediction of *L. plantarum* growth without biotin supplementation, it does not explain how *L. plantarum* synthesizes biotin. Since it is known that biotin is essential for fatty acid synthesis in lactic acid bacteria (34), further investigation of the biotin biosynthesis pathway is needed. The opposite is observed for riboflavin, where experimentally determined auxotrophy is not predicted by iLP728, because riboflavin is not included in the biomass reaction. The core3 and all the original models include riboflavin in the biomass reaction (Table S5), which is wrongly predicted to be synthesized by CarveMe and AGORA models, while core3, gapseq, and modelSEED models do not include its biosynthesis pathway. The other two prediction discrepancies between the core3 and

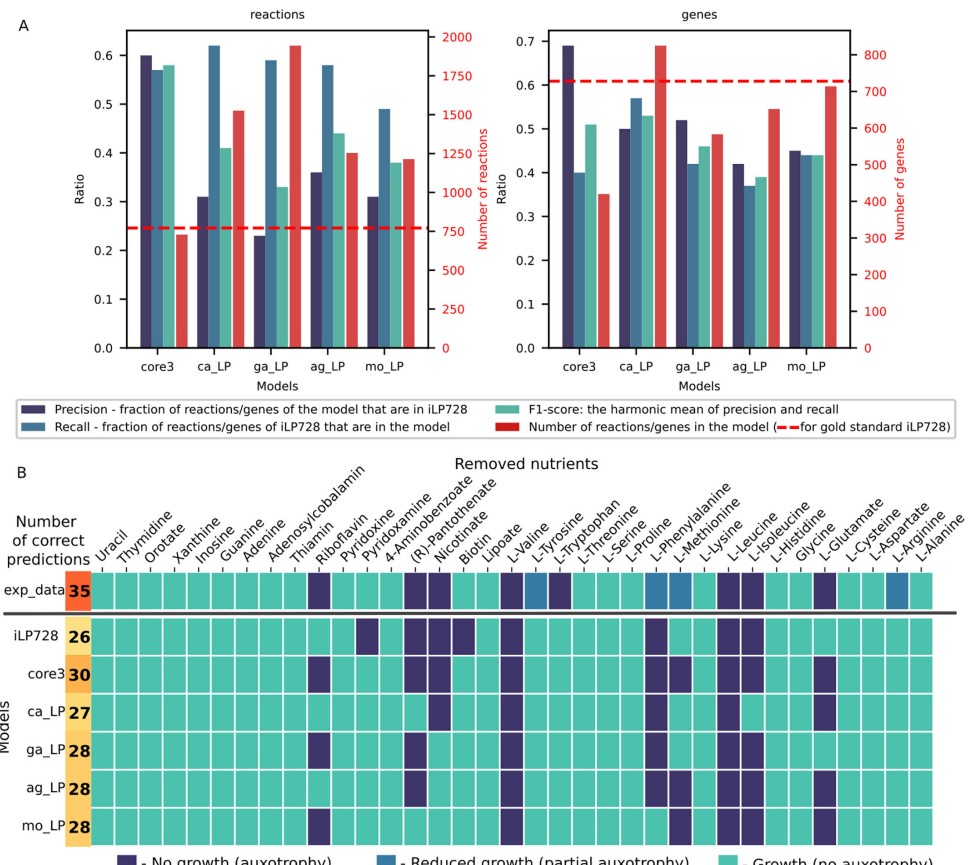

**FIG 4** Core3 *L. plantarum* model curated with GEMsembler outperforms the other models in auxotrophy prediction. (A) Comparison of reactions and genes included in the core3 model, the four original models, and the gold-standard iLP728 model of *L. plantarum*. (B) Prediction of auxotrophy to different nutrients in CDPM medium for all models compared to the experimental information. core3—core3 model curated with GEMsembler; ag_LP—AGORA, ca_LP—CarveMe, ga_LP—gapseq, mo_LP—modelSEED models of *L. plantarum*, exp_data—experimental data.

iLP728 models can be explained by the difference in biosynthesis pathways. In the case of pyridoxamine, the iLP728 model requires it to produce a biomass component, pyridoxal 5′-phosphate, via the ALATA_Lr reaction that does not have a GPR rule (File S10). Core3 model, on the contrary, includes a different pathway to produce pyridoxal 5′-phosphate without pyridoxamine via PYDXS reaction (File S11), which is included in three original models without a GPR rule, therefore correctly predicts that pyridoxamine is not essential. Finally, glutamate auxotrophy is not predicted by iLP728 due to the presence of the P5CD reaction, which enables glutamate production from proline included in the CDPM medium (File S12). This reaction does not have a GPR in iLP728 and is not included by any of the original models; therefore, it is not included in core3, which leads to the correct prediction of glutamate auxotrophy.

In this section, we demonstrated that the model assembly and curation pipeline of GEMsembler, based on the agreement of the original models, produces metabolic models that can accurately describe experimental phenotypes and even outperform the gold-standard curated model. Furthermore, the GEMsembler curated core3 model is the closest in terms of included reactions and genes to iLP728, while being the smallest of all tested models, providing a balance between model complexity and functional capacities. If additional data become available, it can be further curated with the help of GEMsembler, which provides candidate reactions for modification based on their confidence.

## Curating GPR rules with GEMsembler improves gene essentiality predictions in *E. coli*

Model quality depends not only on the network topology, determined by the reactions, but also on the genes included in the GPR rules. GPR rules can be tested by performing gene essentiality predictions, that is, simulating growth while a certain gene is knocked out and the corresponding reactions cannot carry flux, and comparing the results to the experimental data. Gene essentiality predictions can vary between different input models, as the GPR rules can differ, although the same input genome is used to construct the models. Since GEMsembler provides the GPR information from all the original and consensus models, we aimed to investigate whether the space of possible GPRs can be leveraged to improve gene essentiality prediction by combining GPR rules from different models.

For our analysis, we selected the core3 GEMsembler curated *E. coli* model, and the original AGORA, CarveMe, gapseq, and modelSEED models, converted by GEMsembler to BiGG nomenclature and gap-filled with the CarveMe tool on M9 medium. First, as for *L. plantarum,* we checked how similar the curated core3 model and the original models are to the latest curated *E. coli* model iML1515 (18) (Fig. 5A). We note that this version of iML1515 was modified by Bernstein et al. from the original iML1515 model built for the MG1655 strain (33) to account for the differences between MG1655 and BW25113 strain for which experimental gene essentiality data are available. For *E. coli,* the original CarveMe model is the closest one to iML1515, which is not surprising since the universal CarveMe model used for reconstruction incorporates reactions from BiGG *E. coli* models. Core3, while having the smallest number of reactions and the corresponding genes, has the highest precision for genes and the second highest precision for reactions overlapping with iML1515 (Fig. 5A; Table S10). This is also reflected by principal component analysis performed on reaction matrices, as the curated Core3 model is clustering the closest to iML1515 compared to the other models (Fig. S5).

To assess the models' functionality, we compared their gene essentiality predictions with experimental data on fitness defects of 3,789 gene knock-out mutants in 41 minimal media, with 28 and 13 carbon and nitrogen sources, respectively (41, 42). This experimental data set was previously used to evaluate four published curated models of *E. coli* and manually modify the latest curated model, iML1515, to further improve gene essentiality predictions and provide an adjusted model, iML1515a (18). In our analysis, we used 15 carbon sources and all 13 nitrogen sources, on which all models can grow, and calculated the area under the precision-recall curves (AUCPR) for gene-condition pairs sorted by the experimentally determined growth defect as the quality metric (Fig. 5B). The AGORA model with AUCPR = 0.642 outperformed all tested models apart from the adjusted iML1515a model with AUCPR = 0.754, including the standard curated iML1515 with AUCPR = 0.593 (Fig. 5C). It was closely followed by the core3 GEMsembler curated model with AUCPR = 0.556, while the original gapseq, CarveMe, and modelSEED models perform worse with AUCPR between 0.3 and 0.5 (Fig. 5C).

Wrong predictions occur either due to the wrong network topology or the wrong GPR rules. We next leveraged consensus models to address the second issue for each tested model, including iML1515 and its adjusted version. As a first approach, we implemented a stepwise combination algorithm (SA), where for each model, we modified GPR rules that include wrongly predicted genes using GPR rules from the other models ordered by decreasing AUCPR, if the essentiality of the corresponding genes was predicted correctly (Materials and Methods). As a second independent approach, we combined GPR rules from different models with a genetic algorithm (GA), which picks the GPR rules from all possible rules suggested by different models solely based on optimization of the AUCPR for the carbon source conditions. After improving gene essentiality prediction for the data on different carbon sources, we tested whether the improvement was also observed for the nitrogen sources (Fig. 5B).

These two GPR modification algorithms change different numbers of GPRs in the models: while the stepwise algorithm modifies only a few dozens with a minimum of

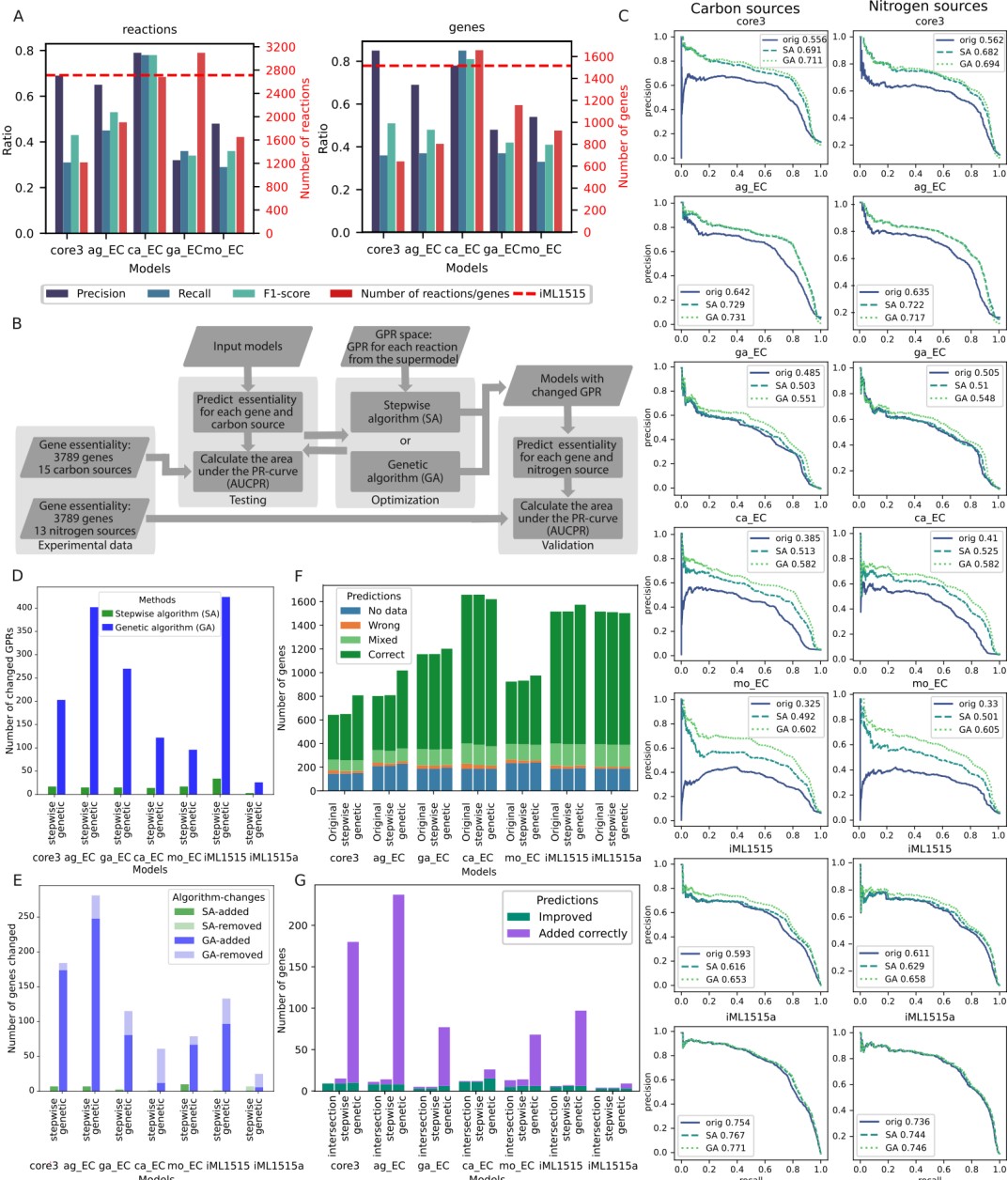

**FIG 5** Combining GPR rules for *E. coli* models improves gene essentiality predictions for growth on different carbon and nitrogen sources. (A) Comparison of reactions and the corresponding genes included in the core3 model, the four original models, and the gold-standard iML1515 model of *E. coli*. (B) Schematic representation of the GPR modification to improve gene essentiality predictions by each model. (C) Precision-recall (PR) curves and their AUC for gene essentiality predictions on carbon sources (left) and nitrogen sources (right) for the original models (blue, solid) and models modified with either the combination algorithm (turquoise, dashed) or the genetic algorithm (green, dotted). (D) Number of changed GPRs in each model by the SA and GA procedures. (E) Number of changed genes in each model by the SA and GA procedures. (F) Number of genes with different prediction status for the original and SA/GA modified models using log2 fold change <−2 for fitness defect compared to the wild type to define no growth. (G) Number of genes for which the essentiality prediction improved using the models modified with either SA or GA, genes with a subset of correct essentiality predictions that were added to the models, and the intersection between the genes improved by SA and GA.

three for iML1515a and a maximum of 34 for iML1515, the genetic algorithm introduces hundreds of changes (Fig. 5D). These changes in GPR may also result in genes being added or removed from the model, with 1–10 genes being altered by the SA and 6–248 genes being altered by the GA (Fig. 5E; Table S10). With the SA, almost no genes were removed from the models except six genes in the adjusted iML1515a model and

one gene in the gapseq model. GA also tends to add new genes rather than remove existing ones, except for the adjusted iML1515a and CarveMe models, which included the highest number of genes originally (Fig. 5E).

Combination of GPR rules with SA improved gene essentiality predictions for all models (Fig. 5C). The performance of the core3 model improved by 13.5% with the final AUCPR = 0.691 which is higher than for the original iML1515 model. The performance of the AGORA model improved by 8.7% reaching AUCPR = 0.729. Gapseq model performance was improved the least by 1.8%, while the performance of CarveMe and modelSEED models improved the most by 12.8% and 16.7% with AUCPR of 0.513 and 0.492, respectively. Even the performance of the gold-standard models could be further improved, reaching AUCPR = 0.616 for iML1515 (2.3% increase) and AUCPR = 0.767 for the adjusted iML1515a (1.3% increase) (Fig. 5C). While it is not surprising that models with a higher prediction quality enhance predictions of the less well-performing models, our results demonstrate that model combination can also be beneficial for the opposite cases. For example, the AGORA model was improved by each of the core3, gapseq, CarveMe, and modelSEED models, which have worse original prediction quality (Table S10).

Using GA to find an optimal GPR combination from the original models improves the gene essentiality predictions for all models even more than SA. The modified core3 and AGORA models reached AUCPR of 0.711 and 0.731, respectively, outperforming the modified iML1515 (Fig. 5C). Gapseq performance improved by 6.6%, while the performance enhancement for the CarveMe and modelSEED models was the largest, with improvement of 19.7% and 27.7% leading to AUCPR of 0.582 and 0.602, respectively. The performance of the gold-standard models could be improved by 6% for iML1515 and by 1.7% for the adjusted iML1515a (Fig. 5C).

Improvement in the overall prediction quality is followed by the increase in the numbers of genes with all correct or mixed predictions for all 15 carbon sources and decrease in the number of genes with the wrong predictions (Fig. 5F). Specifically, 3–15 genes had wrong predictions in different original models, which got improved by either SA or GA procedure (Fig. 5G) across multiple conditions (Fig. S6). In addition, new genes (1–8 genes for SA and 6–229 genes for GA) with correct or mixed predictions were introduced to the models (Fig. 5G, Table S10). There are no genes in any of the models that moved to the wrong prediction category due to SA or GA modifications, and only one gene was added by SA to the ModelSEED model, which had wrong predictions in all 15 carbon sources.

We investigated what genes and GPRs were introduced to the best curated model available to date, the adjusted iML1515a, that improved its performance even further. GA and SA both improved the predictions for three genes *(b0131, b0134,* and *b0778)* that were in the model and introduced one gene *(b1593)* with correct predictions. Each of the improved three genes corresponds to a single reaction in the adjusted iML1515a (ASP1DC, MOHMT, DBTS, respectively). These genes were originally predicted as essential but became non-essential after the GPR modification in agreement with their non-essentiality across all experimental conditions. They became non-essential because the corresponding GPR rules that included only one gene were changed to a GPR rule including two or three isoenzymes. Thus, GEMsembler suggests candidates for taking over the function of these genes. For example, the GPR rule of dethiobiotin synthase reaction (DBTS) was changed from (*b0778*) to (*b0778* or *b1593*). Indeed, *b1593* is annotated in Ensembl (43) as putative dethiobiotin synthetase, which is supported by functional experimental assays (44), and itself has the correct prediction of being non-essential, corresponding to the experimentally determined non-essentiality.

Since it is important to test whether the model improvement can be generalized, we performed gene essentiality predictions for the same mutants grown on different nitrogen sources (Fig. 5C). For the input models and the modified models (with GPR changed based on the carbon sources data), the improvement trends and the overall prediction quality were reproduced in the nitrogen sources data set (Fig. 5C; Table S10).

These results indicate that while experimental data is necessary to automatically improve the models' functional performance, the improved models can also perform better in experimental conditions that were not used for model curation.

Taken together, the GEMsembler framework for comparing different models and building their consensus can be leveraged to combine model features, either manually or automatically, like in the case of gene-protein-reaction rules, to improve the performance of both the original and consensus models.

## DISCUSSION

GEMsembler offers a comprehensive and flexible solution for comparing GEMs built using different tools and assembling their consensus models with different agreement levels. Comparison across tools is essential, since the choice of reconstruction tool can affect the model structure and predictions even more than the input genome. Indeed, it was recently shown for environmental bacterial communities that models built with the same tool were more similar to each other in predicting exchanged metabolites than models built for the same type of community (24). GEMsembler reveals the variability introduced by different tools, provides a functionality to navigate in large metabolic networks, and builds consensus models based on the model agreement, potentially reducing tool-dependent biases. We demonstrate that building consensus models is beneficial because models can improve each other's performance, for example, in auxotrophy and gene essentiality predictions. While experimental data are required to improve model performance, GEMsembler provides (semi)automatic solutions to combine and identify which model features correspond best to the experimental observations, accelerating the design-build-test-learn cycle for cellular metabolism. Furthermore, comparing alternative reactions and GPR rules between different models reveals uncertainties and gaps in our knowledge of the organism's metabolic pathways and helps to prioritize and design experiments to elucidate these gaps. Finally, we propose a systematic and transparent workflow for curating consensus models, demonstrating that the GEMsembler-curated models have similar quality or even outperform the gold-standard curated models of *L. plantarum* and *E. coli*. In this way, GEMsembler can streamline the time-intensive process of model construction and curation while ensuring high-quality models.

GEMsembler also has its limitations, the first being its dependency on BiGG as the primary database, which has fewer metabolites and reactions than, for example, the modelSEED database. We chose BiGG because it is model oriented, widely used in the research community, and has self-explanatory IDs. The flexibility of GEMsembler, however, would allow us to change the primary database, if there is a need in the modeling community. The second limitation is that GEMsembler loses unconverted metabolites, reactions, and genes in the default mode of analysis. We addressed this issue by introducing a mixed option for supermodel generation, which includes non-converted network elements that can be added if they turn out to be important for the analysis. In addition, while GEMsembler currently supports conversion from six model types, the existing model conversion classes can be adapted by the user to new model types.

One of the challenges of consensus model building that GEMsembler attempts to solve is the ambiguous definition of combinations of specific model attributes, such as GPR rules, reaction boundaries, or directionality. For instance, how is an intersection of two GPR rules defined, if in one model the rule is "(A and B) or C" and in another one "(A and D) or E"? On the one hand, it can be A, but on the other hand, it can be empty. We implemented the former option as the default, but included a "and_as_solid" parameter to all functions that include a GPR combination, which can be changed to "True" to choose the latter option. Another combination-related challenge is the numerous ways to combine different models. In this work, we used a rigid approach, which kept model features and their attributes on which a fixed number of models agree. It can result, for example, in cases when a reaction is present because three models

agree on it, but its GPR rule is not retained because there is no overlap between any three models' GPR rules. In the future, we plan to implement a more flexible way to combine feature attributes, such as retaining a GPR rule that corresponds to the highest non-empty model agreement for each reaction. Furthermore, when a pathway from a lower agreement consensus model is added to a higher agreement consensus model, the reactions are currently selected based on pFBA analysis, which means that the added pathway will be minimal and have the overall agreement level of the corresponding consensus model. While the overall agreement of the pathway cannot be increased, since we select it from the consensus model with the highest agreement level in which the pathway exists, an alternative way to select reactions could be to maximize the fraction of reactions with the highest possible agreement within a path (e.g., higher than the overall consensus model), which we consider implementing in the future. Further GEMsembler improvements could include refining the merging of the duplicated reactions, adding metabolite/reaction annotations to the output models, and introducing a possibility for the user to guide the conversion process.

While here we demonstrated that GEMsembler can compare, combine, and aid in the curation of models for single bacterial species, its functionality is not limited to these examples. Alternative pathways identification between different models can inform engineering strategies for strain design to increase the production of metabolites of interest or optimize growth (45). Further applications can include comparing models of closely related species or strains to identify their metabolic differences (23, 46–49). Another application could be comparing different organisms with a similar metabolic phenotype in order to find common pathways that could be responsible for that (50, 51). Furthermore, the use cases of GEMsembler can be expanded to microbial communities (10, 52, 53). An assembly model of the entire community can provide insights into the pathways that are overlapping or complementary between different community members, suggesting potential interspecies interactions and guiding experimental design (54, 55).

Taken together, we believe that GEMsembler's systematic approach to assembling consensus models and ease of use will benefit researchers at any experience level interested in building, analyzing, and curating GEMs of their species of interest. At the same time, GEMsembler's functionality can be generalized and integrated into other computational pipelines addressing various aspects of bacterial metabolism. We therefore hope that GEMsembler will be adopted by a broad scientific community working in the area of systems and computational biology and contribute to building more comprehensive, concise, and biologically informed metabolic models.

## MATERIALS AND METHODS

### GEMsembler package overview and source code availability

GEMsembler package workflow consists of four main steps: (i) input model conversion; (ii) supermodel assembly; (iii) consensus model generation; and (iv) assessment of model agreement and functional analysis. As input, GEMsembler requires model files in SBML format with the information on which tool was used to generate the model. Currently, GEMsembler supports input models built with either of the four tools: CarveMe (10), gapseq (7), MetaNetX (17), and modelSEED (9), or downloaded from either of the two databases: AGORA (13) and BiGG (11). Models from other sources can be implemented by adding a custom model type. Optional input to GEMsmebler, alongside the models, is the bacterial genomes used to generate each of the input models, as well as the genome file or NCBI assembly ID (for automatic file download), which should be used for the final conversion of GPR rules between the models. As output, GEMsembler produces the supermodel that is stored in JSON format, as well as consensus models in the standard SBML format readable by COBRApy (28) or alternative COBRA and model analysis packages (56–58). At the model analysis step, GEMsembler requires the input list with the growth medium components and the metabolites or pathways of interest

and produces plots, tables, and interactive pathway maps in HTML format. GEMsembler is developed in Python and requires standard Python libraries, as well as BLAST (27), which is needed for gene conversion, and the MetQuest package for topological network analysis (29). Out of these dependencies, only BLAST needs to be installed separately by the user. GEMsembler source code, tutorials, and example notebooks are freely available at https://github.com/zimmermann-kogadeeva-group/GEMsembler.

## Model conversion step

In the first step, to convert metabolite IDs of the input models to BiGG IDs (11), GEMsembler uses and prioritizes several sources of crosslink information. We checked for the BiGG IDs provided in the metabolite/reaction annotation field in the input model, as well as crosslink information from the original database of the input model (ModelSEED [8], BiGG [11]). If the IDs provided in the model annotation and the original database overlap, their intersection becomes the first priority for ID mapping. If there are discrepancies between the two, the IDs provided in the model annotation field are considered as the second priority, and if it is empty, the original database crosslinks are considered as the third priority. If there are no conversion results from the model annotation or the corresponding database, the fourth priority is to use some additional conversion information, such as MetaNetX (17) with crosslinks between many databases. If still no BiGG ID is found, the fifth priority is to change the ID according to some pattern (e.g., replace one last underscore with two underscores in AGORA IDs). The last and sixth priority is to check whether the ID itself, without any changes, can be found in the list of BiGG IDs. For each of the currently supported model types, a separate conversion step is implemented. For a custom model type, the user needs to adapt the conversion step according to the model nomenclature. The conversion results between different databases are often ambiguous. One ID in the original database can correspond to several BiGG IDs (one-to-$n$), several different IDs can be converted to one BiGG ($n$-to-one), and even several input IDs can have links to the same set of several BiGG IDs (n-to-n). For different input models that use the same biochemical database nomenclature, such as gapseq and ModelSEED models that both use the ModelSEED database, checking for conversion consistency between models allows eliminating some ambiguous results. For the main track of further steps, GEMsembler selects metabolites to ensure unique one-to-one conversion, and those that do not fulfill that requirement are considered separately.

After the metabolites are converted, the reactions are converted based on their equations. First, only uniquely converted metabolites are used to generate reaction equations in the BiGG nomenclature. For metabolites that were not converted uniquely, for example, one-to-many, GEMsembler uses different possible conversion results to compose potential reaction equations with the corresponding metabolites and automatically checks if any of these equations are present in the BiGG database. If one of the conversion options for a particular metabolite leads to the reaction equations present in the BiGG database, that conversion is used as a conversion result for that metabolite and its reactions. In addition, for models that do not include a periplasmic compartment, this compartment can be introduced if there are reactions for which changing a metabolite compartment from cytosolic or extracellular to periplasmic leads to correct equations present in the BiGG database. In addition, GEMsembler merges reactions with duplicated reaction equations in the models and in the BiGG database, keeping those that are used in the largest number of models.

All stages of conversion are saved in a special "GatheredModels" class, which precedes supermodel assembly. Though only one converted ID, if selected, is used in supermodel assembly, all other potential versions of the converted IDs are available for the user to check in "GatheredModels," including reaction IDs converted only by MetaNetX or other databases, but not supported by reaction equations.

If provided with the input genomes used to generate the corresponding original models, and either an NCBI assembly ID or a genome file in fasta format to be used

for the output models, GEMsembler converts genes included in the input models to the output genome sequences. In case an NCBI assembly ID is provided, the assembly is downloaded automatically according to NCBI ID with the ncbi_genome_download package (59) and locus tags from the assembly are used as gene IDs; otherwise, the model genes are converted to the gene IDs provided by the user in the custom fasta file. Changes introduced by different tools to the gene IDs (like the addition of a dot or the replacement of an underscore), or gene coordinates instead of IDs used by gapseq, are taken into account during the gene conversion process, which can be different for different types of models. The final conversion is made using BLAST sequence alignment (27).

## The supermodel structure and functionality

Information from all the input models is stored in a Python structure called supermodel, with the metadata from all input models available in the "notes" attribute. Supermodel consists of three main classes: metabolites, reactions, and genes, each of which has the following main fields: "assembly," "comparison," and "not_converted." The field "assembly" unites the information on individual metabolites, reactions, or genes from all the input models. These main classes also have fields for each original model for direct access. Not converted metabolites, reactions, or genes are stored in the corresponding "not_converted" field. If the user does not want to lose non-converted elements of the models, it is possible to mix them with the converted ones in the "assembly" using the original non-converted IDs from the input models. Each metabolite, reaction, or gene has similar attributes as in the COBRApy Python class structure, such as "name" or "reactions" for a metabolite, "reactions" for a gene, "metabolites," "genes," "gene_reaction_rule," "lower_bound," "upper_bound" for a reaction, but inside of each attribute, separate fields are added for each input model and their union. These fields allow direct comparison of the values of each attribute in each of the input models and their union.

Supermodel offers various built-in comparison functionalities, for instance, calculating the features on which at least or exactly X input models agree with corresponding "at_least_in" and "exactly_in" supermodel methods. Another supermodel method called "present" allows finding features that are included by a certain list of input models but are excluded by others using logical expressions. Attributes of reactions undergo comparison procedures as well. Lower and upper bounds are selected in such a way that intervals from the input models of interest are intersected, and the resulting intervals are united for possible combinations. Since initially, when assembling a supermodel, we are unsure that all reactions from different models assign reactants and products in the same way, the reversibility of the reaction is determined only by its boundaries. For metabolite coefficients, we made a rule that the coefficient value is assigned to the mode of coefficients from the models of interest, averaged by their possible combinations. Though this rule is arbitrary, it is possible to check for reaction balance and change the coefficient later. If present in the biomass reaction of the original models, the metabolites representing the "Growth-associated maintenance" (GAM), typically integrated in the biomass reaction as an ATP hydrolysis, are treated as any other coefficient. In the same way, the lower bound of the ATP maintenance (ATPM) reaction, the non-growth-associated maintenance (NGAM) term, is treated as any other lower bound in the model. Therefore, if in the original models GAM and NGAM were correctly included in the model, that will propagate to the consensus models. Nevertheless, it is always advised that the user adjusts GAM and NGAM values based on the strain-specific experimental data. For gene reaction rules, we identify parts of the rule on which all models of interest agree, and then unite these parts with "or" for possible model combinations. The results of all performed comparisons are stored in the "comparison" attribute, which is empty at the beginning.

Afterward, it is possible to extract any consensus model, in standard SBML format, as well as add or remove specific reactions the user is interested in by specifying their IDs (if a reaction is added, all its attributes are added from the assembly level). The

consensus level for genes and the biomass reaction can be specified separately. For orphan metabolites in different compartments, additional transport reactions can be integrated. To assist with the stoichiometric balancing of reactions, GEMsembler assesses the mass/charge balance of each reaction with the metabolite formulas from the BiGG database, when available. If the reaction is unbalanced, but can be balanced with coefficients from the original models or by adding/removing hydrogen, the stoichiometry is changed accordingly.

## Analysis of the network topology and function

Once the supermodel is created and the consensus models are built, GEMsembler can be used to perform network topology and functional analysis. For the topology-determined pathway search, we adapted the MetQuest package (29). It needs a path to the XML model file as input, formulation of the growth medium with nutritional sources as a configuration file, and some other optional parameters, for example, a list of metabolites of interest to extract corresponding pathways separately. MetQuest searches for all possible paths below a certain length between the medium components and all metabolites that can be reached using only already available compounds. For an input network with over a thousand metabolites and reactions, such calculations could require >16 GB of RAM (we recommend allocating at least 32 GB of RAM). Therefore, this path search is implemented as a separate "pathsfinding" module, which enables the user to run it on a high-performance cluster, if available. In this work, we ran the "pathsfinding" module using the EMBL Heidelberg HPC cluster (60). As output, the "pathsfinding" module generates a large dictionary with all possible biosynthesis pathways under the certain length in the model in the HDF5 "metquest.h5" file, and a "shortest_paths.pkl" file with a smaller dictionary with three shortest paths (by default) for each metabolite in the model, or metabolites of interest, if specified. Such topological "pathsfinding" outputs for several original and consensus models can be analyzed and summarized together with the "run_metquest_results_analysis" GEMsembler function.

To perform functional analysis of the networks, the FBA and pFBA functions from the COBRApy package are used. Specifically, the culture medium has to be provided to simulate growth and production of all components of the biomass reaction or of metabolites of interest provided by the user. Production of a certain metabolite is considered when the flux through the demand reaction of the specified metabolite, introduced as an objective function for simulations, is higher than 0.001 mmol $g_{DW}^{-1}$ $h^{-1}$. Reactions that carry flux higher than 0.001 mmol $g_{DW}^{-1}$ $h^{-1}$ in the corresponding pFBA results define a biosynthesis pathway for this metabolite. This biosynthesis analysis, together with further summary and analysis for a dictionary of models, is implemented in the "run_growth_full_flux_analysis" GEMsembler function.

Another type of network analysis implemented in GEMsembler with the "get_met_neighborhood" function calculates and visualizes the neighborhood within a number of reactions, which are selected by the user, starting from a given metabolite. This functionality is also used to calculate reactions' distance from the biosynthesis product of the corresponding pathway.

As the output, GEMsember functions "run_metquest_results_analysis" and "run_growth_full_flux_analysis" generate summary production and pathway agreement plots/tables for metabolites of interest in all models. These functions also generate sets of corresponding tables that contain identified paths, as well as interactive network maps built with networkx (version 3.3) (61) and pyvis (version 0.3.2) (62) libraries.

## Input models reconstruction, supermodel, and consensus models generation for *E. coli* and *L. plantarum*

For the use case analysis, the draft models of *L. plantarum* WCFS1 (LP) and *E. coli* BW25113 (EC) were reconstructed with CarveMe and gapseq command-line tools and modelSEED web server. We used protein sequences for CarveMe and modelSEED and nucleotide sequences for gapseq. For *L. plantarum,* we used assembly

GCF_000203855.3, and for *E. coli,* since we reconstructed models from scratch, we used the genome sequence of the strain BW25113, for which gene essentiality data were generated, available from the KEIO collection at https://fit.genomics.lbl.gov/cgi-bin/org.cgi?orgId=Keio. We used gram-specific templates for the models and default gap-filling without specifying any media. We also downloaded models for the species from the AGORA2 collection (13) at https://www.vmh.life/files/reconstructions/AGORA2/version2.01/sbml_files/individual_reconstructions/. As genomes were not available for the AGORA2 collection, we used the genome from AGORA1 (https://www.vmh.life/files/reconstructions/AGORA/genomes/AGORA-Genomes.zip) for *L. plantarum,* and the genome from assembly GCF_000750555.1 for *E. coli*, as it fits to the AGORA2 model in terms of gene IDs, after selecting for locus tags (https://ftp.ncbi.nlm.nih.gov/genomes/all/GCF/000/750/555/GCF_000750555.1_ASM75055v1/GCF_000750555.1_ASM75055v1_cds_from_genomic.fna.gz).

First, for both organisms, the input models were converted with the "GatheredModels" class and its "run" method. Next, two supermodels were assembled with the "assemble_supermodel" method of the "GatheredModels" class: one is the default with "do_mix_conv_notconv" set to False, and the other one is mixed with "do_mix_conv_notconv" set to True parameter. Finally, in both cases, consensus models were created with the "get_all_confident_levels" method of supermodels and the "get_models_with_all_confidence_levels" function. These models were used further for the topological pathway analysis as well as for the growth and pFBA pathway analysis.

As a baseline for model comparison, we took four original AGORA, CarveMe, gapseq, and modelSEED models of *L. plantarum* or *E. coli,* converted to BiGG nomenclature with GEMsembler mixed approach without adding transport reactions, and automatically gap-filled on the minimal PMM5 media with the CarveMe tool gap-filling command. The number of gap-filled reactions added by CarveMe were 8, 0, 3, and 27 for AGORA, CarveMe, gapseq, and modelSEED *L. plantarum*'s models, and 0, 1, 1, and 19 for AGORA, CarveMe, gapseq, and modelSEED *E. coli*'s model, respectively. Note that the CarveMe model for *L. plantarum* and the AGORA model for *E. coli* did not have to be gap-filled because they were able to grow in the tested media.

The biomass components for *E. coli* were taken from the "assembly" field, while for *L. plantarum,* the biomass components were taken from each of the original models and then modified according to the curation procedure below. The reason for the *L. plantarum* difference is that, when using the "assembly" biomass reaction, gap-filling of the original models was not feasible.

## Model curation

The model curation consists of two major steps: (i) curation of the biomass reaction and (ii) curation of the biosynthesis pathways of the biomass components in a given growth medium. To decide which components of the biomass reaction to keep, a preliminary growth analysis and biomass component production analysis were performed using the original models converted to BiGG nomenclature with GEMsembler default and mixed approach. To decide which biomass components to include, we used the agreement score calculated by the GEMsembler "biomass" function to keep biomass components included by three or more models (Table S5). Several metabolites were exceptions to this rule. One of them is acyl carrier protein (ACP_c), which was included by three out of four models, but none of these models could produce it in the preliminary analysis (Table S8); therefore, we decided to exclude it. For *E. coli*, we removed vitamin B12 (adenosylcobalamin adocbl_c), core oligosaccharide lipid Al (colipa_c), and phosphatidy-lethanolamine (dioctadecanoyl, n-C18:0, pe180_c), which were included by three out of four models but not produced by either of them (Tables S5 and S8). We also considered metabolites with agreement by only one or two models, taking into account whether these metabolites can be produced by the corresponding models (Tables S5 and S8). We kept several low-confidence metabolites, such as lipids, because they are essential for biomass production, despite the large disagreements between the models. Two lipid

metabolites and siroheme for *L. plantarum,* as well as one lipid for *E. coli,* were removed because their biosynthesis pathways were long, non-linear, and present only in one model, therefore less likely to be present (Tables S5 and S8). Note that the biomass reaction selection process in GEMsembler is primarily based on the presence of the biomass components in each of the input models and whether production is achieved. Therefore, experimental validation would be necessary for the user to assess the list of biomass components and their corresponding coefficients to obtain a normalized and more accurate biomass reaction for the species of interest.

*L. plantarum* and *E. coli* core3 models were curated to produce all biomass precursors with the following procedure. First, for each biomass precursor that could not be produced by the core3 model, we checked the biosynthesis pathway maps from the core2 model or the original models that can produce the target metabolite. Then, we manually identified a small set of reactions missing in core3 that can restore the metabolite production, and we added these reactions. If, according to the pathway map for the production of one biomass component, another one needs to be produced, the biosynthesis of the latter was curated first. Transport reactions for common metabolites with agreement in less than three models identified during pathway maps exploration were added as well. Finally, if, after adding all reactions from the functional biosynthesis pathway, the curated core3 model was not able to produce the metabolite, we compared reaction boundaries from the original models. For example, for ATP biosynthesis in *L. plantarum,* we compared reaction boundaries between the core3 model and the CarveMe model, which could produce ATP, and found a discrepancy in phosphoribosylaminoimidazole carboxylase (AIRCr) reaction. Changing the reaction boundaries of the AIRCr reaction to bidirectional restored ATP production in the curated core3 model.

MEMOTE quality reports (40) were generated with the command "memote report snapshot." The reports for four original input, four original output, and GEMsembler-curated models per species are available in the git repository at https://git.embl.de/grp-zimmermann-kogadeeva/GEMsembler_paper.

## Comparison with the gold-standard models

As the gold-standard model for *L. plantarum*, we used the original iLP728 model (32). As the gold standard for *E. coli*, we used the iML1515 and iML1515a models published by Bernstein et al., which were adapted by the authors for the BW25113 strain by modifying the original iML1515 model built for the MG1655 strain (33) to account for the differences between the MG1655 and BW25113 strains (18). To assess the overall similarity with the gold-standard models for both *L. plantarum* and *E. coli*, we calculated how much each of the assessed models resembles the gold-standard one (recall) and how much the assessed models are confirmed by the gold-standard one (precision). For reaction-level similarity, we took all reactions from the gold-standard model and each of the assessed ones, and calculated the ratio of the number of intersecting reaction IDs to either the number of all reactions in the gold-standard model (recall) or the number of all reactions in the assessed model (precision). For gene-level similarity, we took a similar approach; however, we only compared genes if they were linked to the same reaction in the compared models (both GPRs from the compared models should contain the gene, but do not have to be the same). Therefore, genes that are included by both compared models but correspond to different reactions were not included in the precision and recall calculation.

Principal component analysis was performed on the reaction matrices to assess model similarity in a low-dimensional space. Reaction matrices were constructed based on the union of all reactions in all compared models (either non-gap-filled GEMsembler-converted original models, consensus models, and the gold standard model; or gap-filled GEMsembler-converted original models, curated Core3 consensus model, and the gold-standard model). Principal component analysis was performed using the PCA

function from the Scikit-learn (version 1.5.1) Python library. Model projections to the first two principal components were plotted.

## Auxotrophy prediction

For auxotrophy prediction in *L. plantarum*, we classified experimental growth in the CDPM media lacking each of the 35 components tested in (34) into three groups: growth ($OD_{600} \geq 1$), no growth ($OD_{600} \leq 0.1$), and reduced growth ($OD_{600}$ between 0.1 and 1) (Fig. 4B). For three metabolites, the growth was modified according to the information provided in a previously published auxotrophy experiment (32). Specifically, the growth without isoleucine was set to zero ($OD_{600}$ of 0 instead of $OD_{600}$ 0.2), because the authors reported that *L. plantarum* cannot produce isoleucine, but growth was observed due to trace elements remaining in the medium (32). Growth without phenylalanine and tyrosine was set to the reduced growth category ($OD_{600} = 0.2$ instead of $OD_{600} = 0.1$), because the authors reported that growth was noticeable in these conditions (32).

To predict auxotrophy with the modeling framework and compare it to the experimental results, FBA was run while optimizing for biomass production as the objective function for each model in each condition where one of the tested nutrients was removed from the CDPM medium. The FBA prediction results were classified into growth [growth rate $\geq 1$ $h^{-1}$ [absolute threshold] or growth rate $\geq 0.85$ of the maximum growth rate calculated in the unmodified CDPM medium of the corresponding model (relative threshold)], and no growth (growth rate $\leq 0.001$ $h^{-1}$ or rate $\leq 0.15$ of the initial maximum growth rate in the unmodified CDPM medium). There were no cases with reduced growth prediction for any of the tested models.

## Gene essentiality prediction

To test the models' ability to predict gene essentiality, we used an experimental data set with the KEIO collection of gene knockout mutants in *E. coli* BW25113 (https://fit.genomics.lbl.gov/cgi-bin/org.cgi?orgId=Keio) (41, 42), grown on minimal media including different carbon and nitrogen sources. This data set was previously compiled to evaluate four published curated models of *E. coli* MG1655, which were modified in the corresponding study to account for the differences between MG1655 and BW25113 strains, and the latest curated model iML1515 was manually modified to improve gene essentiality predictions, resulting in the adjusted iML1515a model (18). For each gene and each condition, a fitness defect measure defined as the log2 fold change of growth compared to the wild type in that condition was available for 3,789 genes in 28 carbon sources and 13 nitrogen sources. We did not need to adjust our models to account for the strain differences since we used the BW25113 genome for model reconstruction; therefore, we directly followed the same gene essentiality prediction pipeline, predicting gene essentiality with COBRApy function "single_gene_deletion" for each of the tested models (four original models, curated core3 model, iML1515, and the adjusted iML1515a from the previous study where they were modified to match the differences in the BW25113 strain). We defined a gene as essential if the predicted growth rate was less than 0.001 $h^{-1}$, and nonessential otherwise. We then sorted the genes according to the experimental fitness defect in increasing order and calculated precision-recall curves and their Area Under the Curve (AUCPR) with Scikit-learn (version 1.5.1) Python library. Note that for each model, we could calculate essentiality for a different number of genes, depending on how many genes from the experimental data set were included in the model. To make the essentiality prediction comparison more fair between the models, we used 15 out of 28 available carbon sources in the data set and 13 nitrogen sources on which all models could grow. For the experimental gene essentiality threshold used to calculate the number of improved predictions, we used a fitness defect of log2 fold change equal to −2, as used in the previous study (18).

## Improving gene essentiality prediction

To improve gene essentiality predictions for each of the tested models, we modified the GPR rules in each model following two approaches. In the first approach, we designed a stepwise GPR rule combination algorithm (SA) to use GPR rules from the other models for modification of the target model. The first step is to select the reactions for which GPR contains a gene with the wrong essentiality prediction in the target model. The next step is to modify the GPR rule by changing it to the GPR rule from a different model, in which the gene essentiality prediction was correct for that gene. If the GPR modification improved the essentiality prediction of the selected gene and did not affect the prediction for the correctly predicted genes by the target model, the GPR was kept. Otherwise, a GPR from a different model for the same reaction was tested. The models that were providing an alternative GPR were tested in the order of decreasing AUCPR.

As a second approach, we implemented a genetic algorithm (GA), which aims to find a GPR rule combination in the solution space that optimizes the AUCPR for gene essentiality prediction of the target model. The reactions of the input model are considered as genes in the chromosome, in the context of the genetic algorithm, if they have different GPR variants in some of the original or core3 models. Potential sources of GPR information were encoded as integers (0 for the input model itself, 1 for core3, 2 for ag_EC, 3 for ga_EC, 4 for ca_EC, and 5 for mo_EC). Selecting one number (one GPR source) per reaction leads to a solution vector, and combining numbers for sources with different GPRs for all reactions forms the solution space for the algorithm. We search for a solution that leads to the highest AUCPR with PyGAD (version 3.4.0) Python package with the following parameters: "num_generations": 50, "num_parents_mating": 40, "sol_per_pop": 200, "parent_selection_type": "tournament," "K_tournament": 20, "keep_elitism": 5, "crossover_type": "two_points," "mutation_type": "random," "mutation_by_replacement": true, "mutation_probability": 0.05, "random_seed": 42, "parallel_processing": 100. The genetic algorithm optimization was executed on the EMBL Heidelberg HPC cluster (60). We also want to note that we encountered numerical accuracy issues when using the glpk solver package when running the "single_gene_deletion" function from the COBRApy package, which we resolved by using the cplex solver instead. To reduce the number of randomly introduced GPR changes, we intersected solutions from different generations, introducing only changes on which the last N generations agree. We started from the last one ($N = 50$), and tested the intersections of the GPRs with the previous generations in terms of the AUCPR, adding one previous generation at a time. The procedure was stopped when the resulting AUCPR decreased below the AUCPR of the 50th generation, rounded to the smaller value with second decimal precision.

## ACKNOWLEDGMENTS

We would like to thank members of the Zimmermann-Kogadeeva lab for helpful discussions and feedback. We would like to acknowledge Gleb E. Gavrish for his help in cross-platform integration of GEMsembler, Artemiy Golden for his input on supermodel creation algorithm, Jean-Karim Hériché and Renato Alves from the EMBL Data Science Centre for advice on the implementation of the genetic algorithm, and Santhust Kumar from the EMBL Data Science Centre for advice on pathways identification. We also thank the EMBL IT Services staff for managing and provision of access to the HPC resources.

This work was supported by the European Molecular Biology Laboratory (EMBL), the EMBL International PhD Programme (E.K.M.), the European Research Council (ERC) (MetaboGutModel-101117769), and the European Union's Horizon Europe research and innovation programme (BlueRemediomics-101082304).

E.K.M., B.B., S.B.-V., and M.Z.-K. conceptualized the overall project and goals. E.K.M. and B.B. developed GEMsembler, implemented genetic algorithm, and wrote all scripts used to generate data and figures. E.K.M. and S.B.-V. examined and curated consensus models.

E.K.M. and M.Z.-K. wrote the first draft of the manuscript, and all authors read, edited, and approved the final manuscript.

In the preparation of this research paper, we utilized generative AI tools (ChatGPT 4o and perplexity.ai) to enhance the clarity and coherence of our writing. We carefully reviewed and edited the AI-generated recommendations to maintain the integrity and authenticity of our work. Generative AI tools were not used to produce new content, but only to rephrase human-written text to improve its clarity. ChatGPT 4o was also used to suggest code implementation to generate figures, which was subsequently edited by the authors.

## AUTHOR AFFILIATIONS

[1]Genome Biology Unit, European Molecular Biology Laboratory (EMBL), Heidelberg, Germany

[2]Collaboration for joint PhD degree between EMBL and Heidelberg University, Faculty of Biosciences, Heidelberg, Germany

## AUTHOR ORCIDs

Elena K. Matveishina  http://orcid.org/0000-0003-4641-4906
Bartosz J. Bartmanski  http://orcid.org/0000-0002-0495-9150
Sara Benito-Vaquerizo  http://orcid.org/0000-0003-3936-9676
Maria Zimmermann-Kogadeeva  http://orcid.org/0000-0001-6021-1246

## FUNDING

| Funder | Grant(s) | Author(s) |
| --- | --- | --- |
| European Molecular Biology Laboratory | | Elena K. Matveishina |
| | | Bartosz J. Bartmanski |
| | | Sara Benito-Vaquerizo |
| | | Maria Zimmermann-Kogadeeva |
| European Research Council | MetaboGutModel-101117769 | Maria Zimmermann-Kogadeeva |
| HORIZON EUROPE Framework Programme | BlueRemediomics-101082304 | Sara Benito-Vaquerizo |
| | | Maria Zimmermann-Kogadeeva |

## AUTHOR CONTRIBUTIONS

Elena K. Matveishina, Conceptualization, Data curation, Formal analysis, Investigation, Methodology, Software, Validation, Visualization, Writing – original draft, Writing – review and editing | Bartosz J. Bartmanski, Conceptualization, Formal analysis, Investigation, Methodology, Software, Visualization, Writing – review and editing | Sara Benito-Vaquerizo, Conceptualization, Methodology, Validation, Writing – review and editing | Maria Zimmermann-Kogadeeva, Conceptualization, Funding acquisition, Methodology, Project administration, Resources, Supervision, Visualization, Writing – review and editing

## DATA AVAILABILITY

GEMsembler source code, tutorials, and example notebooks are freely available at https://github.com/zimmermann-kogadeeva-group/GEMsembler, https://grp-zimmermann-kogadeeva.embl-community.io/gemsembler/, and https://git.embl.org/grp-zimmermann-kogadeeva/GEMsembler/-/blob/master/docs/tutorial.ipynb. Code and data required to reproduce results from the study are available at https://git.embl.de/grp-zimmermann-kogadeeva/GEMsembler_paper. Data and tables used in the study are available on Zenodo: https://doi.org/10.5281/zenodo.16529342.

## ADDITIONAL FILES

The following material is available online.

### Supplemental Material

**Supplemental material (mSystems00574-25-s0001.pdf).** Figures S1-S6, Table S7, supplemental model curation protocol, and list of supplemental tables.

**Table S1 (mSystems00574-25-s0002.xlsx).** General statistics on the numbers in categories from different metabolic models.

**Table S2 (mSystems00574-25-s0003.xlsx).** Topology-determined (MetQuest) production of central carbon metabolites and their confidence.

**Table S3 (mSystems00574-25-s0004.xlsx).** Predefined central carbon metabolism pathways and their confidence.

**Table S4 (mSystems00574-25-s0005.xlsx).** Topology-determined (MetQuest) production of biomass components and their confidence.

**Table S5 (mSystems00574-25-s0006.xlsx).** Biomass reaction composition, its confidence and decision on inclusion in the final biomass reaction.

**Table S6 (mSystems00574-25-s0007.xlsx).** Summary of topology-based confidence analysis and identification of uncertainties.

**Table S8 (mSystems00574-25-s0008.xlsx).** Flux-determined (FBA/pFBA) production of biomass components and their confidence.

**Table S9 (mSystems00574-25-s0009.xlsx).** LP models performance and comparison with gold-standard model.

**Table S10 (mSystems00574-25-s0010.xlsx).** EC models performance and comparison with gold-standard models.

### Open Peer Review

**PEER REVIEW HISTORY (review-history.pdf).** An accounting of the reviewer comments and feedback.

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
