## [Reviewer comments · mSystems]

GEMsembler: consensus model assembly and structural comparison of genome-scale metabolic models across tools improve functional performance

Elena Matveishina, Bartosz Bartmanski, Sara Benito-Vaquerizo, and Maria Zimmermann-Kogadeeva

Corresponding Author(s): Maria Zimmermann-Kogadeeva, European Molecular Biology Laboratory

Review Timeline:

Submission Date:	April 17, 2025
Editorial Decision:	May 11, 2025
Revision Received:	July 29, 2025
Accepted:	August 7, 2025

Editor: Pablo Ivan Nikel

Reviewer(s): Disclosure of reviewer identity is with reference to reviewer comments included in decision letter(s). The following individuals involved in review of your submission have agreed to reveal their identity: Daniel Machado (Reviewer #1)

Transaction Report:

DOI: <https://doi.org/10.1128/msystems.00574-25>

Re: mSystems00574-25 (GEMsembler: cross-tool structural comparison and ensemble modelling improve metabolic model performance)

Dear Dr. Maria Zimmermann-Kogadeeva:

Thank you for your submission to mSystems. Your manuscript has received editorial and peer review consideration, and, on the basis of the external review, I am inviting you to submit a revised manuscript that addresses all reviewer's comments.

Revision Guidelines

Sincerely, Pablo

Prof. Pablo Ivan Nikel
Editor
mSystems

Reviewer #1 (Comments for the Author):

The authors introduce a tool called GEMsembler for semi-automated curation of genome-scale metabolic models. The tool allows for integration can comparison of models produced by other tools. Usage of this tool is exemplified with two case-studies, namely *L. plantarum* and *E. coli*. The authors show how they used their tool to improve auxotrophy prediction in *L. plantarum* and gene essentiality prediction in *E. coli*.

Major issues:

- Overall, I think this is an interesting tool that the scientific community will appreciate. It is hard to predict how widely adopted it will be. My main concern is the balance between automation and user input required for this semi-automated curation. My impression is that the tool mostly consists of a Python API with methods that facilitate integration, but the user still needs to do a lot of the work. I think the authors should discuss more clearly how much manual vs automated work is expected for a typical semi-automated curation workflow.

- For instance, in Lines 592-593 say that for certain reactions "GEMsembler suggests...". Are these steps that need manual intervention ?

- The authors use E. coli BW25113 instead of the usual MG1655 (which is used in the gold standard iML1515 model). These are both K12 strains, so probably not too different in terms of metabolism, but I think the authors should at least mention or elaborate on this issue.

- The authors use the term model ensemble in a way that is very different from how it is usually used in the field (such as the ensembleFBA proposed by Jason Papin, or the ensemble modeling implemented in CarveMe), or even the concept of ensemble modeling used in ODE models or Machine Learning. What the authors call ensemble models, would be more appropriately called consensus models (level 1, 2, 3, etc...).

- Please clarify the integration of model components that can be inconsistent between models like: different substrate/product stoichiometries, GAM/NGAM values, reaction reversibility, upper and lower bounds, etc.

- Also, how is the metadata from the original models handled ? Can it be merged into a consensus ?

- Could the non-matching reactions be due to the failed metabolite-based matching? Did the authors also try to match reaction identifiers through MetaNetX or using EC numbers?

- Lines 220-232: i don't understand the difference between uncertain and disagreement. Can you give an example of something that is classified as disagreement but not as uncertain ?

- I think that a PCA plot with the reaction content of all the 9 models (4 original + 4 ensemble + gold standard) would be interesting to compare their similarity (in addition to the plots in Figures 4a and 5a). This would allow to see who is closer to whom.

- Figure 3C. The pFBA analysis returns a minimal pathway to make a given compound (for example thmpp), but not necessarily the most consensual (highest scoring one). Did the authors consider implementing on objective function for maximizing instead the overall agreement ?

- Lines 238-241: Can't you just check the literature and comment if these reactions are correct or not ?

- Fig 3A and 3B: It took me a while to understand why are there are components synthesized by all 4 models that are not in core4. (I guess that it is because the respective pathways in each model contain at least one reaction that are is not core4.) It is not trivial to see that at first sight, maybe this could be explained to the reader

- iLP728 model is 20 years old. A high agreement with this model (in terms of not including reactions not present in this model) is not necessarily a metric for quality, since the amount of functional annotation in databases has certainly expanded in the meantime.

- The improvement of gene essentiality with the SA and GA method is interesting. I suppose that from all the GA populations, only the best solutions are selected, which explains the improvement. But this requires that the "real experimental data" given. So in reality, like with machine learning, you are only fitting the model to the data, which does not prove the model predictions will improve with unseen data. This could be made clearer to the reader.

Minor issues:

- Please change the color background in all purple circles, otherwise it is almost impossible to read the text inside them (Fig. 1C and in the rest of the figures)

- Line 171: AGORA models was not really generated by the pipeline

- Fig 2 and S2: the colors in sequential colormap in the heatmaps are hard to distinguish (especially dark blue and black), consider using a categorical colormap.

- Fig 2A: agreement level would be easier to understand if assembly, core2, core3, core4 were presented sequentially

- Assembly and core1 are sometimes used interchangeably. And although the authors explain they are the same thing, I recommend to pick one of them and be consistent.
- It would be nice if you mention how many reactions were introduced by CarveMe for each model during gap-filling.
- Line 213: You mention 28 central carbon metabolites, do you mean 30?
- Lines 484-488: This paragraph seems out of place here.
- Fig 4A: please clarify what is indicated by the red bars.

Reviewer #1 (Comments for the Author):

The authors introduce a tool called GEMsembler for semi-automated curation of genome-scale metabolic models. The tool allows for integration and comparison of models produced by other tools. Usage of this tool is exemplified with two case-studies, namely *L. plantarum* and *E. coli*. The authors show how they used their tool to improve auxotrophy prediction in *L. plantarum* and gene essentiality prediction in *E. coli*.

Major issues:

- Overall, I think this is an interesting tool that the scientific community will appreciate. It is hard to predict how widely adopted it will be. My main concern is the balance between automation and user input required for this semi-automated curation. My impression is that the tool mostly consists of a Python API with methods that facilitate integration, but the user still needs to do a lot of the work. I think the authors should discuss more clearly how much manual vs automated work is expected for a typical semi-automated curation workflow.

We thank the reviewer for their general positive assessment of our tool. We agree that the user still needs to do a lot of work to produce high quality models that fit the user's experimental data, if available. However, GEMsembler provides a pipeline that accelerates manual curation steps by providing alternative solutions integrated by different model reconstruction tools for the user to choose from, and also integrates automatic procedures for model curation, such optimization of GPR rules if gene essentiality data is available. To clarify which steps require manual intervention in the model curation process with GEMsembler, we now provide a curation pipeline with the indication of manual steps and automatic steps in the Supplementary materials and in <https://grp-zimmermann-kogadeeva.embl-community.io/gemsembler/>, that perform the preparation for manual decisions, as well as a tutorial notebook with code examples <https://git.embl.org/grp-zimmermann-kogadeeva/GEMsembler/-/blob/master/docs/tutorial.ipynb>.

- For instance, in Lines 592-593 say that for certain reactions "GEMsembler suggests...". Are these steps that need manual intervention?

We thank the reviewer for pointing out that it is unclear which steps require manual intervention. The highlighted section describes model conversion steps, which are performed fully automatically in the GEMsembler workflow. We rephrased the corresponding section to clarify that these steps do not require manual intervention.

- The authors use *E. coli* BW25113 instead of the usual MG1655 (which is used in the gold standard iML1515 model). These are both K12 strains, so probably not too different in terms of metabolism, but I think the authors should at least mention or elaborate on this issue.

We used *E. coli* BW25113 instead of the usual MG1655 strain, because the experimental data on gene essentiality is available for the BW25113 strain, and therefore, we do not have to account for the strain difference when comparing our predictions with the data. Indeed, the two strains are very similar with a difference of several genes between each other (Grenier et al. 2014. Genome Announcements <https://doi.org/10.1128/genomea.01038-14>). The iML1515 model, which we used for comparison, is the version provided by (Bernstein et

al. 2023, MSB), which was already adjusted to the differences between the two strains to correspond to the BW25113, as the authors used the same KEIO gene knockout dataset for model improvement. We added clarifications on the use of these strains and models in the main text as well as in three sections of Methods: on input models reconstruction, comparison with the gold-standard models, and gene essentiality prediction.

- The authors use the term model ensemble in a way that is very different from how it is usually used in the field (such as the ensembleFBA proposed by Jason Papin, or the ensemble modeling implemented in CarveMe), or even the concept of ensemble modeling used in ODE models or Machine Learning. What the authors call ensemble models, would be more appropriately called consensus models (level 1, 2, 3, etc...).

We thank the reviewer for this note and we agree that consensus models are a more appropriate term for our analysis than ensemble models. We changed the term in the text accordingly, and updated the paper title to: "GEMsembler: consensus model assembly and structural comparison of genome-scale metabolic models across tools improve functional performance".

- Please clarify the integration of model components that can be inconsistent between models like: different substrate/product stoichiometries, GAM/NGAM values, reaction reversibility, upper and lower bounds, etc.

We thank the reviewer for pointing out that it is not clear. GEMsembler indeed integrates all these attributes, taking into account inconsistencies between the models, though it does not work specifically with GAM/NGAM. We have added an explanation of the integration procedure in the methods section for the supermodel structure and functionality.

- Also, how is the metadata from the original models handled? Can it be merged into a consensus?

We thank the reviewer for this suggestion. In the previous GEMsembler version at the first manuscript submission, we did not handle model metadata. We now added the functionality for merging metadata from the original models in the "nodes" attribute in GEMsembler, and mentioned the metadata handling in the updated methods section on the supermodel structure.

- Could the non-matching reactions be due to the failed metabolite-based matching? Did the authors also try to match reaction identifiers through MetaNetX or using EC numbers?

We did implement converting all reaction IDs with MetaNetX (though not EC numbers, because they are not available in all the automatically reconstructed models we used) in the GEMsembler conversion stage. There are indeed cases when the reaction equation does not give conversion results, while MetaNetX gives some ID candidates. Nevertheless, we have decided to exclude such IDs, because without having correctly converted metabolites, they likely lead to the wrong topology of the network, can interfere with other reactions or may not be connected to anything. The user can still check potential conversion candidates in the "GatheredModels" class, and we consider allowing the user to interact with the conversion procedure in future GEMsembler updates. Currently, we have decided for the

option of creating a mixed supermodel that contains not-converted features, therefore they are not completely lost. We have added the corresponding clarifications in the methods section on the model conversion step and potential development in the discussion.

- Lines 220-232: i don't understand the difference between uncertain and disagreement. Can you give an example of something that is classified as disagreement but not as uncertain?

We thank the reviewer for pointing out this unclear expression. There is no need to distinguish between disagreement and uncertain, because they are used in different contexts. Disagreement between models refers to a concrete numeric score that tells how many models agree. And when we are talking about uncertain, we mean that based on the concrete disagreement we see (ex., around 50/50), we can not decide whether something exists or not, so we are uncertain and suggest prioritising it. We have added the corresponding clarification in the paragraph that reviewer mentions.

- I think that a PCA plot with the reaction content of all the 9 models (4 original + 4 ensemble + gold standard) would be interesting to compare their similarity (in addition to the plots in Figures 4a and 5a). This would allow to see who is closer to whom.

Following the reviewer's suggestion, we have included four PCA plots (figS5) showing the difference of models based on the absence/presence of all the unique reactions found across all the different *E. coli* and *L. plantarum* models. Two PCA plots correspond to the comparison of gap-filled and non-gap-filled models of *E. coli*, and the other two PCA plots correspond to the comparison of gap-filled and non-gap-filled models of *L. plantarum*. Furthermore, the section on model comparison has been updated to incorporate the results of the newly added PCA plots.

- Figure 3C. The pFBA analysis returns a minimal pathway to make a given compound (for example thmpp), but not necessarily the most consensual (highest scoring one). Did the authors consider implementing an objective function for maximizing instead the overall agreement ?

We thank the reviewer for this great suggestion. In the current version, pFBA analysis is run for consensus models with different agreement levels, and the pathway existing in the consensus model with the highest possible agreement is used for curating models with a higher agreement that do not contain the full pathway. In this way, the curated pathway will be minimal and have the highest possible overall agreement. However, there indeed may exist a longer pathway that contains reactions with a higher agreement (e.g. core3 reactions that are included in the core2 model). We will definitely consider implementing an alternative objective function that includes the maximum number of high agreement reactions in the future updates of GEMsembler. We now discuss this possibility in the discussion section in the updated manuscript.

- Lines 238-241: Can't you just check the literature and comment if these reactions are correct or not ?

We thank the reviewer for the suggestion to add literature links to the highlighted uncertain reactions in *E. coli*. We now included the information on whether there is literature evidence

for each of them (two confirmed by the literature, one rather erroneously added by the automatic tools).

- Fig 3A and 3B: It took me a while to understand why are there are components synthesized by all 4 models that are not in core4. (I guess that it is because the respective pathways in each model contain at least one reaction that are is not core4.) It is not trivial to see that at first sight, maybe this could be explained to the reader.

We thank the reviewer for pointing out this matter. Indeed, the reviewer's interpretation is correct, if the paths to synthesise a metabolite in the input models are different, the consensus model might not be able to synthesise it. We added a clarification of this point to the results section describing figure 3A in the revised manuscript.

- iLP728 model is 20 years old. A high agreement with this model (in terms of not including reactions not present in this model) is not necessarily a metric for quality, since the amount of functional annotation in databases has certainly expanded in the meantime.

We agree with the reviewer that higher agreement with the gold-standard model itself does not necessarily mean higher quality. The high overlap between consensus and the gold-standard models might be not that surprising since the reactions included by most models are likely the confident ones that were also present in an older model. Our main point of comparison is the model functionality, where our consensus model outperforms both the gold standard and the automatically built models. We added a note that assessing functional performance is more important to compare the models' quality in the corresponding results section.

- The improvement of gene essentiality with the SA and GA method is interesting. I suppose that from all the GA populations, only the best solutions are selected, which explains the improvement. But this requires that the "real experimental data" given. So in reality, like with machine learning, you are only fitting the model to the data, which does not prove the model predictions will improve with unseen data. This could be made clearer to the reader.

We thank the reviewer for appreciating the section on gene essentiality prediction. Indeed, gene essentiality data is necessary to improve the predictions, however, GEMsembler framework provides an automated way to benefit from features included differently by different tools, such as GPR rules, by combining them and also learning which feature combination corresponds best to the experimental observations, thus improving our understanding of cellular metabolism. We added a sentence underlying it in the discussion. Furthermore, for *E.coli* knockouts, we used growth data on carbon sources to improve the models, and tested their gene essentiality prediction on nitrogen sources, demonstrating that the performance improved as well, and thus models improved based on a subset of experimental observations can perform better for unseen experiments. We added a note on that in the last section of Results.

Minor issues:

- Please change the color background in all purple circles, otherwise it is almost impossible to read the text inside them (Fig. 1C and in the rest of the figures)

We thank the reviewer for pointing that out, and we have changed the text color in all dark circles to white for better visibility.

- Line 171: AGORA models was not really generated by the pipeline

We thank the reviewer for pointing out this inconsistency. We now rephrased the sentence to clarify that AGORA models were downloaded from the collection.

- Fig 2 and S2: the colors in sequential colormap in the heatmaps are hard to distinguish (especially dark blue and black), consider using a categorical colormap.

We thank the reviewer for pointing that out. Though we indeed use categories, they are sequential (for example, we do not want the “not synthesised” category to be closer to “synthesised” than to “absent”). Therefore, we have changed the colourmap in the heatmaps to a different palette that is still sequential, but provides more colour variation and implemented corresponding changes in GEMsembler.

- Fig 2A: agreement level would be easier to understand if assembly, core2, core3, core4 were presented sequentially

We have changed the order of consensus models in the figure 2A accordingly and implemented an option to provide column order in GEMsembler.

- Assembly and core1 are sometimes used interchangeably. And although the authors explain they are the same thing, I recommend to pick one of them and be consistent.

We agree with the reviewer that using one term throughout the text is better. We changed all cases where “core1” was mentioned to “assembly” or rephrased accordingly, since we prefer to use the term “assembly” as we find it more intuitive.

- It would be nice if you mention how many reactions were introduced by CarveMe for each model during gap-filling.

As suggested by the reviewer, we have indicated the number of reactions introduced by CarveMe in each of the models in the Methods section on the input models reconstruction, supermodel and consensus models generation.

- Line 213: You mention 28 central carbon metabolites, do you mean 30?

We thank the reviewer for pointing that out. It is indeed a typo, and we mean 30. We have changed the number correspondingly.

- Lines 484-488: This paragraph seems out of place here.

We rephrased the paragraph to connect it better to the corresponding results section.

- Fig 4A: please clarify what is indicated by the red bars.

We thank the reviewer for pointing that out, and we added the clarification to the bar legend on top of the second axis on the right in revised Figure 4A.

Re: mSystems00574-25R1 (GEMsembler: consensus model assembly and structural comparison of genome-scale metabolic models across tools improve functional performance)

Dear Prof. Zimmermann-Kogadeeva:
Dear Maria:

I am delighted to report that I am accepting your manuscript for publication in mSystems. I am forwarding it to the ASM production staff: your paper will first be checked to make sure all elements meet the technical requirements. ASM staff will contact you if anything needs to be revised before copyediting and production can begin. Otherwise, you will be notified when your proofs are ready to be viewed.

Sincerely, Pablo

Prof. Pablo Ivan Nikel
Editor
mSystems